



# Pathways of ice-wedge degradation in polygonal tundra under different hydrological conditions

Jan Nitzbon[1,2,3], Moritz Langer[1,2], Sebastian Westermann[3], Léo Martin[3], Kjetil Schanke Aas[3], and Julia Boike[1,2]

[1]Alfred Wegener Institute, Helmholtz Centre for Polar and Marine Research, Telegrafenberg A45, 14473 Potsdam, Germany
[2]Humboldt University of Berlin, Geography Department, Unter den Linden 6, 10099 Berlin, Germany
[3]University of Oslo, Department of Geosciences, Sem Sælands vei 1, 0316 Oslo, Norway

*Correspondence to:* J. Nitzbon (jan.nitzbon@awi.de)

**Abstract.** Ice-wedge polygons are common features of lowland tundra in the continuous permafrost zone and prone to rapid degradation through melting of ground ice. There are many inter-related processes involved in ice-wedge thermokarst and it is a major challenge to quantify their influence on the stability of the permafrost underlying the landscape. In this study we used a numerical modelling approach to investigate the degradation of ice-wedges with a focus on the influence of hydrolog-

5 ical conditions. Our study area was Samoylov Island in the Lena River delta of Northern Siberia, for which we had in-situ measurements to evaluate the model. The tailored version of the CryoGrid3 Land Surface Model was capable of simulating the changing micro-topography of polygonal tundra and also regarded lateral fluxes of heat, water, and snow. We demonstrated that the approach is capable of simulating ice-wedge degradation and the associated transition from a low-centred to a high-centred polygonal micro-topography. The model simulations showed ice-wedge degradation under recent climatic con-

10 ditions of the study area, irrespective of hydrological conditions. However, we found that wetter conditions lead to an earlier onset of degradation and cause more rapid ground subsidence. We set our findings in correspondence to observed types of ice-wedge polygons in the study area and hypothesized on remaining discrepancies between modelled and observed ice-wedge thermokarst activity. Our quantitative approach provides a valuable complement to previous, more qualitative and conceptual, descriptions of the possible pathways of ice-wedge polygon evolution. We concluded that our study is a blueprint for inves-

15 tigating thermokarst landforms and marks a step forward in understanding the complex interrelationships between various processes shaping ice-rich permafrost landscapes.





# 1 Introduction

Many landscapes in the terrestrial Arctic are changing rapidly due to the thawing of ice-rich permafrost and subsequent ground subsidence, a process referred to as thermokarst (French, 2007; Rowland et al., 2010). Thermokarst activity is apparent from various landforms including the formation of lakes, thaw slumps and thermo-erosional gullies (Kokelj and Jorgenson, 2013;

Olefeldt et al., 2016). Another manifestation of thermokarst is the degradation of ice-wedge polygons, which is expressed by the formation of water-filled pits and troughs in polygonal tundra. Advanced degradation of ice-wedges can lead to a transition in the surface micro-topography, from low-centred polygons to high-centred polygons (French, 2007). There is evidence from across almost all of the Arctic for increasing thermokarst activity in general (Kokelj and Jorgenson, 2013) and ice-wedge degradation in particular (Liljedahl et al., 2016) over the past few decades.

Ice-rich thermokarst landscapes are estimated to cover about $20\%$ of the northern hemisphere's permafrost region (Olefeldt et al., 2016) but to store about half of the below-ground organic carbon of that region (Olefeldt et al., 2016). Because ice-wedge polygonal tundra is widespread in thermokarst landscapes it is of particular relevance to the local, regional, and possibly global, cycling of energy, water, and carbon (Muster et al., 2012). The water and energy balances of polygonal tundra are characterized by small-scale spatial heterogeneities (Langer et al., 2011b), which are influenced by the micro-topography of the terrain. The

degradation of ice-wedges and concomitant micro-topographic changes could therefore significantly alter water and energy fluxes in polygonal tundra (Liljedahl et al., 2016), with implications for a range of ecosystem functions such as, for example, the decomposition of organic matter (Lara et al., 2015). However, the implications of ice-wedge degradation for land-atmosphere and land-land energy, water and carbon fluxes remain poorly constrained.

Ice-wedge degradation has been documented through in-situ measurements at, and remote-sensing observations over, a

20 number of different Arctic areas (Jorgenson et al., 2006; Kokelj et al., 2014; Liljedahl et al., 2016; Fraser et al., 2018), and the biogeophysical processes controlling the evolution of ice-wedge polygons on different temporal scales have been described by conceptual models (Jorgenson et al., 2015; Kanevskiy et al., 2017). However, the prediction of thermokarst landscape evolution (as demanded by Rowland and Coon (2015)) requires numerical models capable of simulating the dominant processes associated with thermokarst landforms such as ice-wedge polygonal tundra (Painter et al., 2013). A variety of numerical mod-

25 elling studies have addressed different aspects of a broad range of biogeophysical processes associated with the evolution of ice-wedge polygons. Cresto Aleina et al. (2013) presented a data-driven, scalable approach to assessing the hydrological connectivity of polygonal tundra and reported a non-linear hydrological control on methane fluxes. Several studies have noted the influence of micro-topography and lateral fluxes on subsurface thermal and hydrological regimes, as well as on the biogeochemical processes in polygonal tundra areas (Kumar et al., 2016; Grant et al., 2017a, b; Bisht et al., 2018; Abolt et al., 2018).

Liljedahl et al. (2016) investigated the hydrological implications of a transition from low-centred polygons to high-centred polygons and noted the important influence of spatially heterogeneous snow distribution. Abolt et al. (2017) investigated the ability of a hillslope diffusion approach to describe the geomorphological transition from low-centred polygons to high-centred polygons.





While all of these studies have improved our understanding of certain aspects of polygonal tundra systems, none used numerical models that could simulate the thermal and hydrological processes in polygonal tundra in combination with its dynamically changing micro-topography due to melting of excess ground ice and successive ground subsidence (i.e., thermokarst formation). Furthermore, most of numerical models previously used to investigate ice-wedge polygons involved two- or three-dimensional fine-scale representations of the terrain, thereby significantly limiting the spatial features (a few meters across) and temporal periods (a few years) that could be simulated.

In this paper we present a modelling approach that is able to represent the landscape dynamics of polygonal tundra, thereby taking into account transient micro-topographic changes due to subsidence of ice-rich ground, as well as accounting for lateral fluxes of heat, water and snow. Using a tiling approach to represent different parts of polygonal tundra allowed long-term (several decades) simulations on a landscape scale (several polygonal structures). We took advantage of a detailed in-situ data record from a site in the Lena River delta of Northern Siberia (Boike et al., 2018), which was used to provide model parameters and meteorological forcing data, as well as for comparisons between actual measurements and the simulation results. Our main objectives were:

1. To demonstrate the ability of our tile-based modelling approach to simulate the process of ice-wedge degradation and the associated transition from low-centred polygons to high-centred polygons.

2. To assess the stability and the evolution of ice-wedges in the study area under present-day climatic conditions, but different site-specific hydrological conditions.

3. To quantify the implications of ice-wedge degradation for land-atmosphere and lateral (land-land) water and energy fluxes.

For this we performed and analyzed numerical simulations, using a tailored version of the CryoGrid3 Land Surface Model (Westermann et al., 2016). Our investigations aimed at understanding the the evolution of ice-wedges in polygonal tundra in equilibrium with the recent climatic conditions, while we did not address projections of ice-wedge development in a changing future climate.

## 2 Methods

### 2.1 Study area

Our study area was Samoylov Island in the Lena River delta, Northern Siberia (N 72.36972°, E 126.47500°) which lies within the lowland tundra vegetation zone and is underlain by continuous permafrost. The climate on the island is Arctic continental with a mean annual air temperature (MAAT) below $-12\,°C$, minimum temperatures below $-45\,°C$, and maximum temperatures above $25\,°C$ (Boike et al., 2013, 2018). The average annual liquid precipitation is approximately $169\,mm$. There is typically continuous snow cover from the end of September to the end of May, with a mean end-of-winter thickness of about $0.3\,m$; the



remaining months are mostly snow-free. The permafrost in the Lena River delta extends to depths of between $400$ and $600\,\mathrm{m}$ (Grigoriev, 1960) with ground temperatures (in 2006) at a depth of $27\,\mathrm{m}$ of about $-9\,°\mathrm{C}$ (2006) (Boike et al., 2018).

Samoylov Island belongs to the first river terrace of the Lena River delta, which was formed by fluvial erosion and sedimentation during the Holocene. The sediments in the upper soil layers have a silty to sandy texture with variable mineral ($20$-$40\,\%$ 5 by volume) and organic contents ($5-10\,\%$ by volume) (Boike et al., 2013). Peat layers of variable thickness with higher organic contentents have accumulated at the surface. These Holocene deposits are super-saturated with ice, with an average volumetric ice content of at least $60-70\,\%$ (Zubrzycki et al., 2013). The spatial distribution of ground ice is highly variable due to the presence of syngenetic ice-wedges underlying most of the island. The presence of ice-wedges is expressed at the ground surface by a polygonal-pattered landscape, formed as a result of repeated frost-cracking of the ground and subsequent infiltration 10 and freezing of water. Apart from a few large thermokarst lakes, Samoylov Island is largely covered by ice-wedge polygons whose surface features vary greatly across the island (Fig. 1). Some areas are covered with un-degraded low-centred polygons (LCPs; Fig. 1, D), while initial degradation features, as described by Liljedahl et al. (2016) and Kanevskiy et al. (2017), are visible in other parts of the island (Fig. 1, B). Due to advanced degradation of ice-wedges, some polygons feature a reversed, high-centred relief (high-centred polygons: HCPs) and inter-connected troughs that can be either dry or water-filled (Fig. 1, A 15 and C). The polygonal tundra on Samoylov Island has been previously investigated in a number of studies, with a particular focus on measuring the its water and energy balances (Boike et al., 2008; Langer et al., 2011a, b; Helbig et al., 2013). Moisture levels on the island are spatially variable with a high abundance of polygonal ponds and a few larger water bodies in its central part and dryer, well-drained areas towards the margins of the island (Muster et al., 2012).

The diversity of ice-wedge polygons that evolved under identical climatic conditions on Samoylov Island makes it a highly 20 suitable location for studying the factors affecting the evolution of polygonal tundra. The long-term monitoring of meteorological and ground conditions (Boike et al., 2013, 2018) also provides valuable in-situ baseline data (see Fig. 1 for the location of long-term measurement station). These data have been used as input for permafrost models in a number of modelling studies (Westermann et al., 2016; Langer et al., 2016; Westermann et al., 2017; Gouttevin et al., 2018), as well as for their validation.

## 2.2 Model description

25 ### 2.2.1 Tile-based representation of polygonal tundra

The subgrid-scale heterogeneity of land surfaces (e.g. with regard to vegetation or snow cover) has previously been taken into account in LSMs using tiling approaches (Avissar and Pielke, 1989; Koster and Suarez, 1992; Aas et al., 2017). In this study we have used a similar approach to represent the spatially heterogeneous micro-topography of permafrost landscapes, with each landscape tile representing a distinct part of the polygonal tundra (Fig. 2, A). We subdivided the patterned landscape into 30 three landscape units according to its micro-topography: polygon centres (C), elevated rims (R), and a network of troughs (T) that spreads between the distinct polygonal structures (Fig. 2, B). A similar partitioning of polygonal tundra has been used previously (Muster et al., 2012; Kumar et al., 2016; Grant et al., 2017a, b). Each of these landscape units constitutes a certain *areal fraction* $\gamma$ of the overall landscape. We further simplified the partitioned landscape by assuming that it was made up of





equally-sized hexagons arranged on a regular grid (Fig. 2, C). This assumption allowed us to derive topological relationships between the different landscape units, in particular lateral *distances D* and *contact lengths L* (see Appendix C for details and equations). Each of the landscape units (C, R, and T) was integrated into a single representative tile (Fig. 2, D). Note that together these tiles represented entire areas of the landscape consisting of multiple polygons, and not just a single polygon. The

landscape tiles were coupled by lateral transport processes whose magnitudes were determined by the topological relationships between the relevant tiles (Sect. 2.2.4).

In order to reflect micro-topographic differences within the polygonal tundra landscape, each tile was associated with a particular surface altitude $(a)$.[1] We defined three types of ice-wedge polygonal tundra based on the soil surface altitudes of the tiles. We designated the state in which the centre tile had the lowest elevation as *low-centred polygonal tundra* (LCP):

LCP:     $a_C \leq \min(a_R, a_T)$       .                                                      (1)

Initial ice-wedge degradation is typically characterized by subsidence of the soil surface within the troughs. Those configurations in which the troughs subsided below the level of the centre, but with the rims remaining elevated relative to the centre were designated *intermediate-centred polygonal tundra* (ICP):

ICP:     $a_T < a_C < a_R$       .                                                      (2)

Finally, configurations in which the centre tile had the highest elevation were designated *high-centred polygonal tundra*:

HCP:     $a_C \geq \max(a_R, a_T)$       .                                                      (3)

These definitions of polygonal tundra micro-topography (LCP→ICP→HCP) correspond approximately to the three stages of ice-wedge degradation qualitatively depicted by Liljedahl et al. (2016) in their Fig. 1c, and to the definitions of polygon types by MacKay (2000).[2]

**2.2.2   CryoGrid3 Xice Land Surface Model**

To simulate the ground thermal regime of polygonal tundra, we used a parallelized version of the CryoGrid3 Land Surface Model in which each of the tiles described in Sect.2.2.1 (C, R, and T) was assigned a one-dimensional representation of the subsurface (see Westermann et al. (2016) for a detailed description of the one-dimensional model). This numerical model simulates the temporal evolution of ground temperatures by solving the one-dimensional heat conduction equation, taking

into account the phase change of water through an effective heat capacity. The lower boundary condition is prescribed by a constant geothermal heat flux $(Q_{geo})$, while the boundary condition at the surface is given by the ground heat flux $(Q_g)$, which is obtained by explicitly solving the surface energy balance (SEB) equations from the meteorological forcing data. The model further simulates the build-up and ablation of the snowpack, heat conduction within the snow, and water infiltration and refreezing within the snowpack.

---

[1]Throughout this paper we used the term altitude $(a)$ when referring to the height above sea level, and the term elevation $(e)$ when referring to the height above the initial position of the centre tile $(a_C)$ which serves as a reference height.

[2]Note, that we excluded the case $a_R < a_C < a_T$ from these definitions as it corresponds to a state that is not observed in nature.





The unique feature of CryoGrid3 that enables it to simulate the evolution of ice-wedge polygons is its excess ground ice scheme ("Xice"), which uses a simple parameterization for excess ice melt and the resulting ground subsidence and water body formation, based on an algorithm proposed by Lee et al. (2014). This involves that each cell of the discrete one-dimensional grid that represents the subsurface in the model, is assigned a "natural" porosity ($\phi_{\mathrm{nat}}$). Frozen grid cells for which the volumetric

water content ($\theta_{\mathrm{w}}$) exceeds $\phi_{\mathrm{nat}}$ are considered to contain excess ice. Once a grid cell containing excess ice thaws, the excess water ($\theta_{\mathrm{w}} - \phi_{\mathrm{nat}}$) is routed upwards while the solid soil matrix material of the cells above is routed downwards such that it occupies the "natural" volumetric matrix fraction ($1 - \phi_{\mathrm{nat}}$). Continued thawing of excess ice cells results in a net subsidence of the soil surface and potentially in the formation of a water body at the surface, depending on the treatment of excess water.

### 2.2.3   Hydrology scheme for unfrozen ground

To simulate the subsurface hydrological regime of the active layer we enhanced CryoGrid3 with a hydrology scheme for unfrozen ground conditions. We used an instantaneous infiltration scheme which assumes rapid vertical water flow compared to the rates of other processes represented in the model. This is a valid assumption for the upper soil layers of tundra wetlands, which are typically characterized by large hydraulic conductivities (Boike et al., 2008) and in which infiltration within the active layer is dominantly controlled by its thickness (Zhang et al., 2010).

Given the preconditions of a snow-free and unfrozen soil surface, the hydrology scheme simulates the change in water content ($\theta_{\mathrm{w}}$) of each grid cell (within the soil domain and water bodies atop the soil surface) resulting from precipitation, evapotranspiration and infiltration (Fig. 3). Water gained from rainfall (and snowmelt) is added to the uppermost soil grid cell. Reductions in soil water content due to evaporation and transpiration are distributed down to a characteristic evaporation depth ($d_{\mathrm{E}}$) and root depth ($d_{\mathrm{T}}$). An infiltration algorithm is then used to take into account the changes in water content due to

precipitation and evapotranspiration; this first routes water downwards to the bottom of the active layer, setting the soil water content of all grid cells at maximum to the field capacity ($\theta_{\mathrm{fc}}$). Potential excess water is then pooled upwards by successively saturating the grid cells, which leads to the formation of a water table within the active layer (or even above the soil surface). A quantitative description of the hydrology scheme is given in Appendix A.

### 2.2.4   Lateral transport of heat, water, and snow between tiles

We further enhanced CryoGrid3 by taking into account lateral subsurface heat and water fluxes between the landscape tiles, as well as snow redistribution. Topological characteristics of and relationships between the tiles (area $A$, thermal distance $D^{\mathrm{th}}$, hydraulic distance $D^{\mathrm{hy}}$, contact length $L$) were used to quantify the magnitudes of the lateral fluxes.

*Lateral transport of heat:* The lateral heat flux between adjacent tiles is computed for each grid cell $i$ of all tiles, according to Fourier's law. The heat flux $q^{\mathrm{th}}_{\alpha,i}$ [$\mathrm{J\,s^{-1}}$] to cell $i$ of tile $\alpha$ from all adjacent tiles is given as

$$30 \quad q^{\mathrm{th}}_{\alpha,i} = \sum_{\beta \in \mathcal{N}(\alpha)} k^i_{\alpha\beta} \frac{T^i_\beta - T^i_\alpha}{D^{\mathrm{th}}_{\alpha\beta}} L_{\alpha\beta} \Delta^i_\alpha \,, \tag{4}$$





where $\mathcal{N}(\alpha)$ denotes all tiles adjacent to $\alpha$, $k^i_{\alpha\beta}$ is the effective lateral thermal conductivity (Eq. (B1) in Appendix B), $T^i$ refers to the temperature and $\Delta^i$ to the height of grid cell $i$. The lateral heat flux is added to the vertical heat fluxes resulting from heat conduction and boundary fluxes (i.e., geothermal and ground heat fluxes).

*Lateral transport of water between tiles:* Lateral water fluxes between adjacent tiles are calculated as bulk fluxes, based on Darcy's law. Given the precondition of a snow-free and unfrozen land surface, the lateral water flux $q^{\mathrm{hy}}_\alpha$ [m s$^{-1}$] to tile $\alpha$ from all adjacent tiles is given as

$$q^{\mathrm{hy}}_\alpha = \sum_{\beta \in \mathcal{N}(\alpha)} K_{\alpha\beta} \frac{w_\beta - \max(w_\alpha, f_\alpha)}{D^{\mathrm{hy}}_{\alpha\beta}} \frac{H_{\alpha\beta} L_{\alpha\beta}}{A_\alpha} , \tag{5}$$

where $K_{\alpha\beta}$ is the saturated hydraulic conductivity between tiles $\alpha$ and $\beta$, $w_\alpha$ the elevation of the water table of tile $\alpha$, and $f_\alpha$ the elevation of the frost table (i.e., the bottom of the active layer) of tile $\alpha$. $H_{\alpha\beta}$ is the hydraulic contact height between tiles $\alpha$ and $\beta$ which is obtained as follows:

$$H_{\alpha\beta} = \min\left[w_\beta - \max(w_\alpha, f_\alpha), w_\beta - f_\beta\right] . \tag{6}$$

Note that a tile for which no water table forms above the frost table, can only receive lateral water fluxes. To ensure conservation of water, lateral fluxes are proportionally reduced if insufficient water is available in those tiles that "lose" water.

*Exchange of water with the surrounding terrain:* The trough tile is hydrologically connected to a theoretical external "water reservoir" that has a constant water level ($w_{\mathrm{res}}$).[3] This water level serves as a hydrological boundary condition between the modelled polygonal tundra and the surrounding terrain. A low water level in the external water reservoir leads to drainage of the troughs while a high water level results in their inundation. The lateral water fluxes from the reservoir to the troughs ($q^{\mathrm{hy}}_{\mathrm{res}}$) are calculated in a similar way to those in Eq. (5), as follows:

$$q^{\mathrm{hy}}_{\mathrm{res}} = K_{\mathrm{res}} \frac{(w_{\mathrm{res}} - \max[w_{\mathrm{T}}, f_{\mathrm{T}}])^2}{A_{\mathrm{T}}} \tag{7}$$

where $K_{\mathrm{res}}$ is the reservoir hydraulic conductivity that, compared to the saturated hydraulic conductivity $K$, also incorporates the topological parameters for hydraulic distance ($D^{\mathrm{hy}}$) and contact length ($L$) between the reservoir and the trough tile.

*Lateral transport of snow:* Snow redistribution due to wind drift is assumed to occur between all tiles. A terrain index ($I_\alpha$) that depends on the difference of the snow surface elevation ($a_\alpha$) and the mean surface elevation ($\bar{a}$) is calculated, and indicates whether tile $\alpha$ loses or gains snow due to snow redistribution. The mobile snow of the more elevated tiles is redistributed amongst the less elevated tiles while at the same time taking into account the conservation of mass, leading to a levelling out of the snow surfaces, The "snow catch" effect of vegetation is taken into account by treating only snow above a threshold height ($h^{\mathrm{catch}}_\alpha$) as "mobile" snow. Furthermore, lateral snow transport does not occur during melting conditions, i.e., if any snow cell has a positive temperature ($T^i > 0$) or contains liquid water. The governing equations of the lateral snow transport scheme are provided in Appendix B.

---

[3]Note that we used the symbol $e_{\mathrm{res}}$ when the reservoir water level is given relative to the initial altitude of the centre tile ($e_{\mathrm{res}} = w_{\mathrm{res}} - a_{\mathrm{C}}$).



### 2.3 Model setup and simulations

#### 2.3.1 Topology

As described in Sect. 2.2.1, we represented the spatial heterogeneity of the polygonal tundra landscape using three tiles (C, R, and T) and a theoretical, external water reservoir (denoted with subscript "res"). Figure 4 provides a schematic representation

of the lateral connections between these tiles and indicates important parameters used to characterize the topology and micro-topography in the model setup. Polygon centres are adjacent only to the rims, while the rims are also connected to the troughs. The troughs are hydrologically connected to the external water reservoir, which makes it possible to exchange water with the surrounding terrain (Fig. 4). We derived the topological relationships between the tiles (areas, contact lengths, and distances) based on an assumed regular hexagonal grid (see Fig. 2, C). We estimated the areal fraction ($\gamma$) of the tiles (Table 1) on the

basis of land cover classifications for Samoylov Island by Muster et al. (2012). The contact lengths ($L_{\alpha\beta}$), thermal distances ($D^{\text{th}}$), and hydraulic distances ($D^{\text{hy}}$) between adjacent tiles were calculated on the basis of geometric assumptions (i.e., regular hexagonal grid) and the mean size of polygons on Samoylov Island given by Muster et al. (2012) (see Fig. 4 and Appendix C for details).

#### 2.3.2 Micro-topography

The lateral fluxes of snow, water, and heat are also influenced by the micro-topography of the terrain, which is reflected in the different relative elevation of the tiles. We assumed that the initial elevation of the rims ($e_{\text{R}}$) of intact low-centred polygons relative to the centres was $0.4\,\text{m}$ (Boike et al., 2018), and that the troughs were initially $0.1\,\text{m}$ lower than the polygon rims (Table 1). The water level in the external water reservoir relative to the initial altitude of the polygon centres ($e_{\text{res}}$) was varied between different runs. In order to take into account variability in the polygon topology and micro-topography we conducted a

sensitivity test and compared the modelled results with in-situ measurements (see Sect. 3).

#### 2.3.3 Parameters

Many of our model parameters were adopted from previous studies of the same study area that also used CryoGrid3 (see Table D1 in Appendix D). The parameters introduced with the hydrology scheme (Sect. 2.2.3) and the lateral transport scheme (Sect. 2.2.4) are summarized in Table 2, which also includes the default values and ranges assumed in this study. We assumed a field

capacity for the upper soil layers ($\theta_{\text{fc}}$) of $0.50$, which is in agreement with typical volumetric water contents in unsaturated soil layers, measured during the summer (Boike et al., 2018, Table 1). The root depth ($d_{\text{T}}$) was set to $0.2\,\text{m}$ for the polygon centres and troughs, which are typically covered with mosses and sedges, and to $0.1\,\text{m}$ for the rims, where the mosses have shallower roots. The evaporation depth ($d_{\text{E}}$) was set uniformly to $0.1\,\text{m}$ for all tiles. The saturated soil hydraulic conductivity ($K$) between all connected tiles was set to $1 \cdot 10^{-5}\,\text{m}\,\text{s}^{-1}$ and the reservoir hydraulic conductivity ($K_{\text{res}}$) was set to $5 \cdot 10^{-5}\,\text{m}\,\text{s}^{-1}$; both values

were of the same order of magnitude as the various estimates for the same study area provided in Boike et al. (2018). The lateral



transport schemes were run in intervals of six hours ($\Delta t_{\text{lat}} = 0.25\,\text{day}$). Those parameters for which no published values were available have been estimated by the authors who have long-time field experience in the study area.

### 2.3.4 Soil stratigraphies

The soil stratigraphies were based on the stratigraphy provided in (Westermann et al., 2016) for a polygon centre on Samoylov Island. However, we modified the stratigraphies for the different tiles to reflect the spatially heterogeneous ground ice distribution, which is linked to the surface micro-topography expressed as polygon centres, rims and troughs (see Table 3). The position of excess ground ice layers was crucial for the geomorphological dynamics simulated by the subsidence scheme. In these layers the volumetric water content exceeds the natural porosity ($\phi_{\text{nat}}$), for which we assumed a conservative value of $0.55$. In an idealized polygonal tundra, ice-wedges are located beneath the troughs, where frost cracks occur during cold winters. We assumed that the ice-wedges consisted of almost pure ice ($\theta_w^0 = 0.90$, i.e. $35\%$ excess ice), and that they extended from a depth of $0.5\,\text{m}$ down to $9.3\,\text{m}$. An intermediate layer with less excess ice ($\theta_w^0 = 0.75$, i.e. $20\%$ excess ice) was placed between $0.2\,\text{m}$ and $0.5\,\text{m}$ depth, serving as a "protective" layer between the active layer and the ice-wedge (cf. the conceptual model by Kanevskiy et al. (2017)). Above these excess ice layers, the troughs were covered with an insulating organic soil layer $0.2\,\text{m}$ thick. Since ice-wedges typically extend laterally beneath the polygon rims (which is causal for their elevation above polygon centres), we assigned high excess ice contents to the rim tile over the full vertical extent of the ice-wedges. We therefore placed an excess ice layer with $\theta_w^0 = 0.75$ (i.e. $20\%$ excess ice) between $0.6\,\text{m}$ and $9.4\,\text{m}$ depth. A silty mineral layer was placed above that excess ice layer, covered by an organic-rich layer $0.1\,\text{m}$ thick. The stratigraphy for the polygon centres was chosen to be identical to that in Westermann et al. (2016), with a layer of $10\%$ excess ice starting at a depth of $0.9\,\text{m}$ and extending downwards to $9.0\,\text{m}$. Note that the excess ice domain of all tiles ends at the same absolute depth and that this depth corresponds to the recorded depth of ice-wedges on Samoylov Island. The upper organic layer of the centre tile had a thickness of $0.15\,\text{m}$. A bedrock layer with no organic constituents and a lower ice content of ($\theta_w^0$) of $0.30$ was assumed to extend from the bottom of the ice-rich layer down to the lower end of the model domain for all tiles (C, R, and T).

### 2.3.5 Forcing data

The model required meteorological data (including air temperature, humidity, pressure, wind speed, rain and snow precipitation, and incoming long-wave and short-wave radiation) to provide the atmospheric forcing at the upper boundary of the model domain. We used the same forcing dataset as Westermann et al. (2016), which covers the period from 1979 to 2014. These data are based on in-situ observations from Samoylov Island for the period from 2002 to 2014 (Boike et al., 2018). Downscaled ERA-Interim reanalysis data were used to infill gaps during this period and to obtain forcing data for the period from 1979 to 2002 (see Westermann et al. (2016) for details). In order to allow long-term simulations under present-day climatic conditions the dataset was extended to 2040 by repeated appending of the forcing data from the period between 01/2000 and 12/2014 to the end of the original forcing dataset. Note that due to a lack of consistent in-situ data the precipitation forcing (rain and snow) was taken from the reanalysis product for the entire forcing period.



### 2.3.6 Simulations

An overview of the different model runs and specific parameter settings is provided in Table 4.

*Validation runs:* We conducted eight 7-year validation runs for the period from 10/2007 to 12/2014, for which there is a good coverage of in-situ data available. All runs were initialized with a typical temperature profile for the beginning of

5 October, based on borehole measurements from 2006 and an extrapolation assuming a typical geothermal temperature gradient ($0\,\mathrm{m}$ depth: $0.0\,°\mathrm{C}$, $2\,\mathrm{m}$: $-2.0\,°\mathrm{C}$, $5\,\mathrm{m}$: $-7.0\,°\mathrm{C}$, $10\,\mathrm{m}$: $-9.0\,°\mathrm{C}$, $25\,\mathrm{m}$: $-9.0\,°\mathrm{C}$, $100\,\mathrm{m}$: $-8.0\,°\mathrm{C}$, $1100\,\mathrm{m}$: $+10.2\,°\mathrm{C}$). The initial soil water content of the active layer was based on the end-of-season soil moisture measured for the centre and rim profiles in 2007 (Boike et al., 2018). We allowed the state variables to adjust to the climatic conditions during an entire winter season before comparing model output with measurements. We confirmed that this spin-up period was sufficient for the near-surface

processes of interest by evaluating the same period after a 10-year spin-up with the meteorological forcing of 2007, but this did not change the evaluation results significantly. We used a number of validation runs to test the model's sensitivity to variations in micro-topography between different polygons. For this we varied the areal fractions of the centre tile ($\gamma_\mathrm{C}$) and the rim tile ($\gamma_\mathrm{R}$); the areal fraction of the trough tile ($\gamma_\mathrm{T}$) was 0.1 for all runs. We also varied the elevations of the rims ($e_\mathrm{R}$) within realistic ranges, as well as the snow density ($\rho_\mathrm{snow}$), as this is known to exert a critical influence on the soil thermal state (Table 4,

VALIDATION). We then compared the spread of the results from the eight validation runs with in-situ measurements (see Sect. 3).

*Long-term runs:* To study the long-term (i.e., over multiple decades) evolution of polygonal tundra under different hydrological conditions, we conducted a number of 60-year runs for the period from 10/1979 to 12/2039. The soil temperatures were initialized to the same profile as used for the validation runs. The initial soil water contents were as shown in Table 3. The first

three months (10/1979-12/1979) of the simulations were not considered in order to allow the state variables to adjust to the climatic forcing. We confirmed that this spin-up period was sufficient by running the model with a 20-year spin-up using the meteorological forcing of the 1979-1989 decade, which did not change significantly the results for the analysis period starting from 1980. The lateral topology and micro-topography of the polygonal tundra were estimated based on available in-situ data given and referenced in Table 1. To investigate the susceptibility of the evolution of polygonal tundra to the hydrological condi-

tions, the elevation of the external water reservoir ($e_\mathrm{res}$) was varied between $-1.5\,\mathrm{m}$ and $+0.5\,\mathrm{m}$ (Table 4, LONGTERM-XICE), where low values correspond to drainage of the troughs and higher values to their inundation. To isolate the effects of subsidence we conducted control runs with the subsidence due to excess ice melt disabled (Table 4, LONGTERM-CONTROL). For these runs the initial LCP micro-topography was static during the entire simulation period.

## 3 Model validation

In order to justify the tile-based representation of a permafrost landscape within our modelling framework we assessed its ability to reproduce the spatial heterogeneity in the thermal and hydrological characteristics of polygonal tundra by comparing the model results with in-situ measurements from Samoylov Island.





## 3.1 In-situ measurements

The in-situ measurement data came mainly from the long-term records of the Samoylov Permafrost Observatory described in
Boike et al. (2018). This dataset contains vertical soil temperature and soil moisture profiles of different micro-topographic
units of the polygonal tundra (see Fig. 1 for the location of the measurement station), as well as water table (WT) records

for a polygon centre. We also used the active layer thickness (ALT) time series from the Samoylov Circumpolar Active Layer
Monitoring (CALM) site which cover different micro-topographic units of polygonal tundra, including polygon centres ("wet
tundra") and rims ("dry tundra") (Boike et al., 2013). Most of the data cover the period from 2002 to 2017 (with a few gaps),
but WT measurements are only available from 2007 to 2017. In order to evaluate the simulated spatial heterogeneity of the
surface energy balance (SEB) of polygonal tundra, we took surface energy flux measurements and Bowen ratios recorded in

summer 2008 from different parts of Samoylov Island by (Langer et al., 2011b). We used spatially distributed snow depth
(SD) measurements obtained in spring 2008 from different parts of the polygonal tundra, including polygon rims and polygon
centres (Boike et al., 2013), for comparison with the modelled spatial distribution of snow.

## 3.2 Comparison of model results and in-situ measurements

### 3.2.1 Ground thermal regime

We compared the modelled and measured evolution of ALT for polygon centres and rims during all years of the validation
period, i.e. from 2008 to 2014 (Fig. 5: left for centres, right for rims). For the polygon centres both the progression of the thaw
front and the maximum ALT of all model runs lay mostly within one standard deviation from the mean of the measurements.
During the last four years of the validation period (2011-2014) the ALT progression in all model runs showed a large variability
and some runs simulated ALTs that were too shallow compared to the measured values. For 2013 all model runs significantly

underestimated the maximum ALT (by about $0.3\,\mathrm{m}$) compared to the measured values. This can probably be attributed to too
little precipitation in the forcing data, leading to an overly dry, and hence insulating, upper organic soil layer.

The modelled ALT progressions for polygon rims were within the range of available measurements during all years of
the validation period, resulting in good agreement with respect to both the timing of thawing and the maximum ALT. The
underestimation of ALT for the centres that occurred in 2013 was not observed for the rims. There was generally a very

low variability between the validation runs, indicating that the modelled ALT for the rims was less susceptible to changes in
the varied parameters ($\gamma_C$, $e_R$, $\rho_{\mathrm{snow}}$) than that for centres. This can probably be attributed to larger differences between the
hydrological regimes of the centers in the different validation runs.

In addition to the ALT, we also compared modelled and measured soil temperatures within the active layer, for both polygon
centres and polygon rims (see Fig. E1 in Appendix E).



### 3.2.2 Ground hydrological regime

The hydrology scheme incorporated into CryoGrid3 (see Sect. 2.2.3) made it possible to simulate a dynamic water table; we compared the modelling results with in-situ WT measurements from a polygon on Samoylov Island (see Fig. 1 for the location of the measurement polygon). Figure 6 shows the modelled and measured WT evolution in the polygon centre during the
summer months of the validation period. With the exception of 2013, the simulated WT evolutions showed a large range of about $0.1$ to $0.2\,\mathrm{m}$ between the most extreme runs. For some years (e.g., 2011) there were runs in which the centres were water-covered throughout the entire summer while for other runs the water table was mostly below the soil surface. This suggests that the simulated WT is highly susceptible to the topology (i.e. the areal fractions $\gamma$) and micro-topography (i.e. the elevation of rims $e_\mathrm{R}$) of the polygonal tundra. There was a general pattern in the simulated WT evolutions that was independent of the year
and the parameter setting used: WTs were high immediately after snowmelt, usually followed by a decrease over the summer months after which the WTs stabilized towards the end of summer or increased again in response to major precipitation events.

    The measured WTs lay mostly within the range of the simulated WTs. During most summers the measured WTs were partly above and partly below the soil surface. In most years the measured WT evolutions revealed a similar pattern to the modelled evolutions, with high WTs after snowmelt followed by a decrease towards the end of the summer. However, the measurements
showed a more pronounced intra-annual variability than the individual model runs. In contrast, the inter-annual appeared to be slightly greater in the model runs. The remaining mismatches between modelling results and measurements were attributed to a variety of factors including (i) site-specific characteristics of the measurement polygon, (ii) the accuracy of the precipitation forcing data used in the model, (iii) the accuracy of measured WT values, and (iv) the simplistic representation of vertical and lateral water fluxes in the model.

Apart from WT we also compared modelled and measured soil moisture levels within the active layer, for both polygon centres and polygon rims (see Fig. E2 in Appendix E).

### 3.2.3 Summer surface energy balance

We used eddy-covariance measurements from Samoylov Island obtained between 7 June and 30 August, 2008 (Langer et al., 2011b) to assess the model's ability to reproduce the spatial heterogeneity in the SEB of polygonal tundra. We first compared
measured mean surface energy fluxes[4] (net radiation $Q_\mathrm{net}$, sensible heat flux $Q_\mathrm{h}$, latent heat flux $Q_\mathrm{e}$, ground heat flux $Q_\mathrm{g}$) with the modelled area-weighted mean fluxes for the same period in the validation runs (Fig. 7, left hand side). The variability in all SEB components between the different model runs was low and they were all close to, or within, the uncertainty range of the measured fluxes. The measured summer Bowen ratio ($B_\mathrm{meas}$) was 0.50, which compares well with the mean modelled value ($\bar{B}_\mathrm{model}$) of 0.58. Note that the low variability in the modelled SEB partitioning indicates that it is robust against variations in
the topology and micro-topography of the polygonal tundra.

    We next compared the turbulent heat fluxes ($Q_\mathrm{h}$, $Q_\mathrm{e}$) of wet tundra estimated from field measurements with those modelled for the centre tile (Fig. 7, central part). While the modelled fluxes lay within the large uncertainty range of the fluxes measured

---

[4]The eddy-covariance measurement footprint was about $200\,\mathrm{m}$ in diameter.



in the field, the mean modelled summer Bowen ratio ($\bar{B}_{\mathrm{model}}^{\mathrm{wet}}$) of 0.39 was larger than the measured value ($B_{\mathrm{meas}}^{\mathrm{wet}}$) of 0.02. However, the SEB partitioning for the centre tile in the model was significantly distinct from the area-weighted mean SEB.

Similarly, the modelled turbulent heat fluxes for the rim tile were of comparable magnitude to the fluxes for dry tundra estimated from field measurements (Fig. 7, right hand side). The mean summer Bowen ratio for the rims in the model runs ($\bar{B}_{\mathrm{model}}^{\mathrm{dry}}$) was 0.87, while the measured value ($B_{\mathrm{meas}}^{\mathrm{dry}}$) was 1.29. Like for the wet tundra described above, the model was able to reproduce a SEB partitioning for the rim tile that was distinct from the area-weighted mean SEB.

Although the spatial heterogeneity was more pronounced in the measured data than in the modelling results, the model was able to reproduce spatially heterogeneous patterns of turbulent heat fluxes while also robustly reflecting the mean summer SEB partitioning of the polygonal tundra.

### 3.2.4 Snow redistribution

To assess the ability of the snow redistribution scheme to reproduce actual, spatially heterogeneous, SD distributions, we compared modelled and measured SDs for spring 2008 in polygon centres, rims, as well as the areal mean SD (Fig. 8). The mean modelled SD for polygon centres, in all validation runs ($\bar{\mathrm{SD}}_{\mathrm{model}}^{\mathrm{C}}$) was about $0.51\,\mathrm{m}$ and thus lay within the range of the measurements, which had a mean ($\bar{\mathrm{SD}}_{\mathrm{meas}}^{\mathrm{C}}$) of about $0.46\,\mathrm{m}$. Similarly, the mean modelled SD of the rims ($\bar{\mathrm{SD}}_{\mathrm{model}}^{\mathrm{R}}$) was about $0.21\,\mathrm{m}$ and again lay within the measurement range, which in this case had a slightly lower mean ($\bar{\mathrm{SD}}_{\mathrm{meas}}^{\mathrm{R}}$) of about $0.18\,\mathrm{m}$. The modelled and measured areal mean SDs ($\bar{\mathrm{SD}}_{\mathrm{model}}$ and $\bar{\mathrm{SD}}_{\mathrm{meas}}$, respectively) were almost identical and amounted to $0.31\,\mathrm{m}$. The comparison showed that the model was able to realistically reproduce the spatial heterogeneity in SD. Furthermore, the variability within the validation runs with different micro-topographies, areal fractions, and snow densities, was similar to that found in the measurement data collected from an area that contained a number of polygons.

### 3.3 Summary

The comparison of modelled and measured ALT, WT, SEB and SD (plus soil temperature and soil moisture in Appendix E) justified the use of a tile-based modelling approach as the measured spatial heterogeneities in (i) the subsurface thermal and hydrological regimes, (ii) the surface energy balance, and (iii) the snow distribution of polygonal tundra, were well reproduced in the model validation runs.

## 4 Results

### 4.1 Ice-wedge degradation under intermediate hydrological conditions ($e_{\mathrm{res}} = 0.0\,\mathrm{m}$)

The primary objective of our study was to simulate the transient process of ice-wedge degradation in a tile-based model of polygonal tundra. For this we conducted 60-year runs of our model setup with enabled excess ice module (see Table 4, LONGTERM-XICE). The run with an intermediate water level in the external water reservoir ($e_{\mathrm{res}}$) of $0.0\,\mathrm{m}$ illustrated very





well the degradation process and the associated micro-topographic changes (Fig. 9). The supplementary material to this article contains an animated video showing the results of this simulation run.

*LCP phase:* During the first decade of the simulation period (1980-1990) the modelled polygonal tundra was was low-centred (LCP, according to definition (1) above). During this period the centre-tile was (over-)saturated with water, resulting in

surface water of $0.05$ to $0.2\,\mathrm{m}$ depth. The rims were rather dry with end-of-summer WTs about $0.2 - 0.3\,\mathrm{m}$ below the surface. The absolute altitude of the modelled end-of-summer WT was almost identical for the centre (1986: $\mathrm{WT_C} = 20.20\,\mathrm{m}$) and rim tile (1986: $\mathrm{WT_R} = 20.15\,\mathrm{m}$), indicating lateral water fluxes that lead to a leveling out of the WT within the active layer between the different tiles. During the LCP phase the troughs showed a very shallow ALT of about $0.2\,\mathrm{m}$ depth and their active layer was mostly dry with water tables either absent or just above the frost table. Note that during the first nine years of the simulation

period the active layer of the trough tile did not extend below the water level in the external water reservoir ($20.0\,\mathrm{m}$ a.s.l.), so that any water in the troughs drained into the external reservoir.

*Transition:* During the last five years of the LCP phase (1986 to 1990) the troughs started to subside as soon as the active layer extended into the intermediate excess ice layer, which lay between $0.2$ and $0.5\,\mathrm{m}$ depth and contained $20\%$ excess ice (see Table 3). Concurrent to the subsidence of the ground surface the end-of-summer ALT almost doubled from $0.20\,\mathrm{m}$ in 1986

to $0.38\,\mathrm{m}$ in 1991. During these years the water saturation in the active layer beneath the troughs increased (with the WT lying above ALT) due to the addition of water from the melting of excess ice and lateral fluxes from the rims, which occur every summer as soon as the frost table of the troughs sinks below the water table of the rims. The increased amount of liquid water in the active layer beneath the troughs increased its thermal conductivity, resulting in increased ground heat flux (with the mean summer $Q_\mathrm{g}$ in the trough tile increasing from about $8\,\mathrm{W\,m^{-2}}$ in 1986 to about $36\,\mathrm{W\,m^{-2}}$ in 1991) which in turn resulted in a

further increase in the ALT. This positive feedback led to continued melting of excess ice and subsequent ground subsidence in the following years. In year 1989 of the simulation the thaw front extended into the ice-rich layer representing the ice-wedge which contained $35\%$ of excess ice (see Table 3). The increased amount of melted excess ice pooled up in the active layer and enhanced the positive feedback described above.

*ICP:* During summer 1990 of the simulation the soil altitude of the troughs subsided below that of the centre tile, such

that the polygons represented by the tiles were classified as ICPs, according to definition 2) above. After the first two years of the ICP phase (1990-1991), during which the troughs subsided very rapidly (about $0.2 - 0.3\,\mathrm{m\,a^{-1}}$), the subsidence rate decreased to about $0.05 - 0.1\,\mathrm{m\,a^{-1}}$ in subsequent years, with a number of summers recording no ground subsidence at all. The total subsidence of the troughs between 1990 and 2012 amounted to about $1.0\,\mathrm{m}$. The depth of the water body ponded in the troughs was about $1.0\,\mathrm{m}$ in 2012. During the ICP phase the rims also started to subside as soon as the active layer extended

into the excess ice layer for the first time (extending downwards from $0.6\,\mathrm{m}$ depth with $20\%$ excess ice, see Table 3). The subsidence continued for about two decades but at a lower rate than the troughs, which had a higher excess ice content. During the simulation period from 1990 to 2012 the polygon rims subsided by a total of $0.4\,\mathrm{m}$, reaching the level of the centre tile, so that the modelled polygonal tundra was subsequently high-centered (HCP, according to definition (3 above). Note that during the ICP phase the active layer of the polygon rims became wetter as the absolute altitude of the WT remained more or less





constant while the soil surface subsided. The polygon centres remained mostly water-saturated during the ICP phase, with their WT sinking below the soil surface in only a few of the summers.

*HCP:* The HCP phase lasted from 2012 until the end of the simulation period in 2040. The subsidence rates for both rim and trough tiles were significantly lower than during the ICP phase. No ground subsidence occured in any of the tiles during the
last decade of the simulation. The water levels in all tiles also stabilized and showed less inter-annual variability. As soon as the centres became the highest part of the landscape their WT dropped to about $0.2\,\mathrm{m}$ below the surface. The resulting organic-rich dry upper layer had an insulating effect so that the maximum ALT of the centre tile was substantially lower during the HCP phase than during the preceding LCP and ICP phases.

All of the stages of polygonal tundra evolution defined in Sect. 2.2.1 occurred during the 60-year simulation period of the run
with an intermediate water level in the external water reservoir ($e_{\mathrm{res}} = 0.0\,\mathrm{m}$): the LCP phase lasted for 10 years, from the start of the simulation in 1980 until the summer of 1990, followed by the ICP phase which lasted for 22 years (until summer 2012) and the HCP phase which continued for the remaining 27 years. The key characteristics of the different phases are summarized in Fig. 10, which shows boxplots of the distributions of end-of-summer ALT, end-of-summer WT, and maximum SD for each tile, together with the area-weighted means for the entire landscape. The transition from LCP to HCP led to an increase in the
maximum ALT for polygon rims and troughs but a reduced maximum ALT for the polygon centers centres. These changes were directly linked to the changes in WTs which fell from above the surface of the centres during the LCP phase to below the surface during the HCP phase. For the polygon rims and troughs the WT increased such that the landscape-mean WT elevation relative to the soil surface increased slightly, indicating that the polygonal tundra was becoming wetter. The snow distribution, which was heterogeneous during the LCP phase with a high SD for centres and troughs, becomes increasingly homogeneous
during the HCP phase where the surface altitudes of the three tiles were similar.

In summary, the run with intermediate hydrological conditions ($e_{\mathrm{res}} = 0.0\,\mathrm{m}$) demonstrated that the tile-based approach to modelling a polygonal tundra landscape is able to simulate the degradation of ice-wedges and the associated geomorphological transition from low-centred polygons to high-centred polygons.

## 4.2   Variation of the hydrological conditions

The second objective of our study was to investigate the control that hydrological conditions exert on the evolution of polygonal tundra. For this we considered the results of additional long-term runs with different water levels in the external water reservoir ($e_{\mathrm{res}}$; see Table 4, LONGTERM-XICE). To contrast the results for the run with $e_{\mathrm{res}} = 0.0\,\mathrm{m}$ discussed in Sect. 4.1, we analyzed in detail another model run with a rather low value for $e_{\mathrm{res}}$ of $-1.0\,\mathrm{m}$, corresponding to draining hydrological conditions (Sect. 4.2.1). We also compared the evolution of the polygonal tundra in all runs with the excess ice scheme enabled, covering a broad
range of hydrological conditions (4.2.2).

### 4.2.1   Draining hydrological conditions ($e_{\mathrm{res}} = -1.0\,\mathrm{m}$)

The temporal evolution of the tiles is shown in Fig. 11 and the characteristics of the different phases are summarized in Fig. 12. The supplementary material to this article contains an animated video showing the results of this simulation run.




*LCP:* The landscape evolution for this setting can be divided into two phases. During the first three decades of the simulation period (1980-2010) the polygonal tundra remained low-centred, with the polygon centres being water-covered, the rims were stable (i.e. not subsiding) with an end-of-summer WT about $0.3$ m below the surface, and the troughs were dry, draining into the external water reservoir. During this phase the troughs subsided slightly (by $0.05$ to $0.1$ m) due to thawing of excess ice in

the intermediate excess ice layer, which extended from $0.2$ m to $0.5$ m depth (see Table 3). The thawing of excess ice in the trough tile accelerated towards the end of the third decade of the simulation period (between 2007 and 2010). This resulted in an increase in the amount of liquid water in the active layer of the trough tile, which was not compensated by the runoff into the external water reservoir. A positive feedback through increasing thermal conductivities and ground heat fluxes, analogous to that described in Sect. 4.1, was thus initiated, which in turn resulted in sustained ice-wedge degradation over the next two

decades.

*ICP:* The ICP phase start with subsidence of the troughs to below the level of the centre tile in summer 2011, and continued until the end of the simulation period in 2040. The positive feedback described above caused the troughs to continue subsiding despite the drainage of the trough network into the external water reservoir. The SD in the deepening troughs increased from a maximum of about $0.3$ m during the LCP phase to one of about $1.0$ m during the ICP phase. This resulted in increased

liquid water input from snowmelt into the active layer of the troughs, thus enhancing the positive feedback through thermal conductivities and ground heat fluxes. During the ICP phase the water tables receded in both centre and rim tiles. The lower WT in the centres (WT was about $0.2$ m below the surface) compared to the LCP phase (when the centres were mostly water-covered) indicated that the rims, despite their relative elevation ($e_R$) of $0.40$ m, did not prevent the centres from being drained by lateral subsurface water fluxes. With WTs about $0.6$ m below the surface, the active layer of the rim tile was also well-

drained during the ICP phase. While the rims subsided very little (only about $0.05$ m) until year 2025 of the simulation, this was followed by a phase of accelerated excess ice melt with about $0.15$ m of ground subsidence between 2025 and 2030. The landscape stabilized during the last decade of the simulation period (2030-2040), with no subsidence occurring in any of the tiles. Until the end of the 60-year run the rims remained elevated by about $0.2$ m above the centres, so that the HCP phase was not attained for the run with $e_{res} = -1.0$ m.

The changes in ALT, WT and SD associated with the landscape evolution of the run with draining hydrological conditions are summarized in Fig. 12. Although the mean ALT did not change much with the transition from LCP to ICP, the spatial pattern changed substantially, with an increase in ALT for polygon rims and troughs compensated by a marked reduction for the polygon centres. While the LCP micro-topography resulted in isolated, water-saturated centres, their end-of-summer WT fell significantly during the ICP phase to about $0.2$ m below the surface. The end-of-summer WT of the rims also decreased

by about $0.2$ m with degradation of the ice-wedges. The mean water level fell from about $-0.1$ m relative to the soil surface to almost $-0.5$ m, indicating an overall drying of the landscape. The change in snow cover was most pronounced for the troughs. In the drained troughs above the degraded ice-wedges up to about $1.0$ m of snow accumulated during the winters of the ICP phase.





### 4.2.2 Comparison between all runs under different hydrological conditions

The presented results of the two model runs for (i) an intermediate water level in the external water reservoir ($e_\text{res}$) of $0.0\,\text{m}$ (see Sect. 4.1), and (ii) a rather low external water reservoir level of $-1.0\,\text{m}$ (see Sect. 4.2.1) revealed that the hydrological conditions exerted a strong influence on the evolution of the polygonal tundra. While ice-wedge degradation occurred in

both runs, the timing and the speed of this process varied between the runs, as indicated by the timing and duration of the different phases (LCP, ICP, and HCP). Ice-wedge degradation (i.e. the transition from LCP to ICP) in the wetter setting (with $e_\text{res} = 0.0\,\text{m}$) started about two decades earlier (in 1990) than in the dryer setting (with $e_\text{res} = -1.0\,\text{m}$), where it occurred in 2011. The excess ice melt in troughs and rims was more rapid during the wet setting (with $e_\text{res} = 0.0\,\text{m}$), where the ICP phase lasted 22 years, compared to the dry setting (with $e_\text{res} = -1.0\,\text{m}$) where it lasted more than 29 years.

To illustrate the dependency of ice-wedge stability on the hydrological conditions we compared the evolution of polygonal tundra between all runs at different levels of $e_\text{res}$ (see Table 4 for the parameter values and Fig. 13 for the results). Since the transition from the LCP phase to the ICP phase marks the initiation of ice-wedge degradation, its timing is indicative of the stability of the original LCP landscape. For $e_\text{res} > 0.1\,\text{m}$ excess ice melt began immediately after the start of the simulations, such that the transition to the ICP phase occurred within the first two years. For intermediate external water reservoir levels

($e_\text{res} = 0.1\,\text{m}$ and $e_\text{res} = 0.0\,m$) ice-wedge degradation started within the first decade of the simulation period. For all runs with lower water levels in the external water reservoir ($e_\text{res} < 0.0\,m$) the transition from LCP to ICP occurred after about three decades of simulation time in the summer of 2011.

The duration of the ICP phase, which terminates as soon as the the rim tile subsides below the centre tile, can be used as an indicator of the speed of ice-wedge degradation. The ICP phase was generally shorter for those runs with inundating

hydrological conditions (e.g., for $e_\text{res} = 0.5\,\text{m}$ the ICP phase lasted about 15 years) than for the dryer settings (e.g., for $e_\text{res} = -0.5\,\text{m}$ the ICP phase lasted more than 29 years), for which the HCP phase was not adopted until the end of the simulation period. However, for intermediate water levels in the external water reservoir ($e_\text{res} = 0.1\,\text{m}$ and $e_\text{res} = 0.0\,m$) the HCP phase was reached by 2012, one year later than in the run with $e_\text{res} = 0.2\,\text{m}$, meaning a shorter duration for the ICP phase. This is counter to the general trend of slower degradation with lower water levels in the external water reservoir, but can probably be

explained by exceptionally high excess ice melt in the 2010-2012 period, which is about at the time that the ICP phase was reached in the runs with $e_\text{res} \leq -0.1\,\text{m}$.

Hydrological conditions that led to a drainage of the troughs, were generally found to stabilize the landscape and to slow down the melting of excess ground ice. Exceptionally extreme meteorological conditions can, however, initiate or accelerate ice-wedge degradation, irrespective of the hydrological conditions.

## 4.3 Implications of ice-wedge degradation for water and energy fluxes

After investigating the evolution of polygonal tundra under different hydrological conditions, our third objective was to quantify the effect of ice-wedge degradation on the water and energy fluxes in polygonal tundra. We observed significant changes to both land-atmosphere fluxes (reflected by evapotranspiration) and land-land fluxes (reflected by external runoff), induced by the



degradation of ice-wedges and the associated changes in micro-topography. While these fluxes showed a very low sensitivity to the hydrological conditions if a static LCP micro-topography was assumed, ice-wedge degradation was found to increase the susceptibility of the fluxes to the hydrology, but in a non-linear fashion.

### 4.3.1 Evapotranspiration

To investigate the implications of ice-wedge degradation for land-atmosphere fluxes we looked at the differences between accumulated summer (i.e. snow-free period) evapotranspiration (ET) for runs with enabled excess ice module (LONGTERM-XICE) and with the module disabled (LONGTERM-CONTROL); in addition we varied the water level in the external water reservoir (see Table 4). Since the micro-topography remained static when the excess ice module was disabled, the polygonal tundra remained in the LCP phase over the entire simulation period in all runs. For all of the runs with the excess ice module enabled, however, ice-wedge degradation was observed during the 60-year simulation period (i.e. the transition from LCP to ICP occurred; see Fig. 13). In order to exclude the influences of the meteorological forcing and isolate the effect of micro-topographic changes we compared ET during the last 10 years (2030-2039) of the runs with the excess ice module enabled with those when it was disabled (Fig. 14, left panel).

In the control runs the ET showed no significant dependence on the hydrological conditions for low and intermediate $e_{\text{res}}$. For $e_{\text{res}} \leq 0.1\,\text{m}$ the ET ranged between 125 and 150 mm, while for higher water levels in the external reservoir ($e_{\text{res}} \geq 0.2\,\text{m}$) the ET increased up to about 200 mm for $e_{\text{res}} = 0.5\,\text{m}$. Note that the relative elevation of the rims ($e_{\text{R}}$) was 0.4 m, so that external water levels above this level led to an entirely water-covered landscape. We observed significant changes to the ET in runs with the excess ice module enabled, with a non-linear dependence on $e_{\text{res}}$. For runs with low $e_{\text{res}}$ values ($\leq -0.35\,\text{m}$) the ET was by about 10-20 mm lower than in those runs with static micro-topography. For runs with high $e_{\text{res}}$ values ($\geq 0.1\,\text{m}$) the ET increased significantly (i.e. by more than 50 mm) to above 200 mm. For runs with intermediate $e_{\text{res}}$ values the results showed a marked increase in the ET as the $e_{\text{res}}$ increased.

### 4.3.2 Runoff

In conjunction with our investigations into the changes of land-atmosphere fluxes (as reflected in ET), we also investigated changes in the lateral (i.e. land-land) water fluxes between the model domain its surrounding terrain. For this we looked at the accumulated summer (i.e. snow-free period) runoff (R) from the troughs to the external water reservoir. To exclude the effects of the meteorological forcing we again compared the runoff for the same simulation period (2030-2039) between runs with the excess ice module disables (LONGTERM-CONTROL) and the excess ice module enabled (LONGTERM-XICE) (Fig. 14, right).

For the control runs with low and intermediate $e_{\text{res}}$ the external runoff was mostly positive (i.e., net flux from the model domain to the external water reservoir), with mean accumulated annual fluxes in the order of 5 mm during the last 10 years of the simulation period. Only for high water levels in the external water reservoir ($e_{\text{res}} \geq 0.2\,\text{m}$), R decreased to negative values (i.e., net flux from the external water reservoir to the model domain). For most $e_{\text{res}}$ values we observed significant changes to R within the same period, if the micro-topographic changes induced by ice-wedge degradation were taken into account (i.e.





with enabled excess ice module). For $-1.0\,\mathrm{m} \leq e_{\mathrm{res}} \leq -0.35\,\mathrm{m}$ there was a significant increase in R to mean annual values of about $25\,\mathrm{mm}$. For high reservoir levels ($e_{\mathrm{res}} \geq 0.1\,\mathrm{m}$), however, R decreased substantially to mean values of about $-50\,\mathrm{mm}$ if ground subsidence was enabled. There was a sharp reduction in R for intermediate values of $e_{\mathrm{res}}$, yielding an overall non-linear relationship between the two quantities.

It is noteworthy that for $e_{\mathrm{res}} = -1.5\,\mathrm{m}$ R increased only slightly to about $10\,\mathrm{mm}$ if the excess ice module was enabled, while the increase in runoff was larger for higher water levels in the external reservoir (e.g. $e_{\mathrm{res}} = -0.5\,\mathrm{m}$). This could probably be attribute to the slower degradation of the ice-wedges for this very low value of $e_{\mathrm{res}}$ (cf. the results in Sect. 4.2.2), resulting in higher rims during period under consideration which would in turn impede lateral water fluxes from the polygon centres into the troughs. Note that for most $e_{\mathrm{res}}$ values the absolute value of R (i.e. its modulus) was multiple times higher for the
runs with enabled excess ice module than for the runs with static micro-topography. Only for $e_{\mathrm{res}} = -0.1\,\mathrm{m}$ did R not change significantly when ground subsidence was taken into account.

## 5   Discussion

### 5.1   Ice-wedge degradation as a transient phase in the evolution of polygonal tundra

The long-term (60-year) runs with variable hydrological conditions demonstrated the ability of our model framework to reflect
the process of ice-wedge degradation and the associated changes to the micro-topography of polygonal tundra in a realistic way. During the initial years of the two extensively discussed modelling runs (see 4.1 and 4.2.1), the low-centred polygonal tundra prevailed with similar characteristics between the runs regarding the active layer thickness (ALT), water tables (WT) and snow depth (SD) (see LCP phase in Figs. 10 and 12).

The timing of the initiation of ice-wedge degradation, i.e. the time at which the active layer in the trough tile extended
down to the ice-rich layer representing the ice-wedge (see Table 3), was found to depend on the hydrological conditions (reflected in $e_{\mathrm{res}}$), with higher water levels in the external water reservoir leading to an earlier onset of degradation and lower water levels (i.e. drainage conditions) having a stabilizing effect (see Fig. 13). This suggests that ice-wedge degradation is triggered by the hydrological regime in the troughs above the ice-wedges, which results from a combination of the hydrological conditions (i.e. the hydrological "forcing") and the meteorological forcing of the relevant year. As suggested by Kanevskiy et al.
(2017), ice-wedge degradation may therefore be initiated by extreme meteorological conditions in certain years, such as major precipitation events or high air temperatures, which would result in exceptionally large ALTs. This is supported by our finding that the intermediate-centred (ICP) phase started in the same year in all runs with $e_{\mathrm{res}} \leq 0.1\,\mathrm{m}$ (see Fig. 13). Our results suggest that those parts of the polygonal tundra that are well-drained are less susceptible to ice-wedge degradation than wetter parts, irrespective of any meteorological forcing, and that the rate of excess ice melt is lower under dryer conditions.

Once the degradation process was initiated, the simulations showed continuing degradation and ground subsidence of the troughs in subsequent years. This rapid degradation was observed to be independent of the hydrological conditions and to continue for two or three decades, until a new equilibrium state was reached. We suspect that this process is driven by a positive feedback loop, with meltwater resulting from the melting of excess ice being routed upwards and thus increasing





the thermal conductivity within the active layer; this would increase the ground heat flux which would in turn result in an increased ALT, leading to further melting of excess ice. The feedback would be slowed down as the solid soil material stored in the excess ice layers accumulates to a soil layer without excess ice and a new "equilibrium ALT" is established. The presence of ponded water within the troughs for the run with intermediate hydrological conditions ($e_{res} = 0.0\,\mathrm{m}$; Fig. 9) would not

inhibit this process because of the efficient temperature mixing that occurs during summer and the insulating effect of the water body during freeze-back. Since rapid subsidence of the troughs was also observed for draining hydrological conditions ($e_{res} = -1.0\,\mathrm{m}$; Fig. 11), this appears to indicate that the drainage of water from the troughs into the external reservoir is not sufficient to break this feedback loop. Indeed, the lower part of the active layer beneath the troughs was found to be saturated during the ICP phase, even under draining conditions (see Fig. 11, years 2010 to 2020). This fact may relate to increased snow

accumulation in the deepening troughs as a result of lateral transport from the polygon centres and rims, leading to increased amounts of meltwater in the active layer beneath the troughs during spring.

There is a lack of reliable, long-term measurements of ground subsidence for the different micro-topographic units of polygonal tundra in our study area, which makes quantitative comparisons with the modelled subsidence rates unfeasible. However, Boike et al. (2018) reported recent (2013 to 2017) subsidence rates on Samoylov Island to be in the order of $0.04\,\mathrm{m\,a^{-1}}$ for poly-

gon rims and $< 0.01\,\mathrm{m\,a^{-1}}$ for polygon centres. These figures are in agreement with the modelled subsidence characteristics, with rates of about $0.02\,\mathrm{m\,a^{-1}}$ for the rim tiles and no subsidence for the centre tiles (see Figs. 9 and 11).

With the establishment of a new equilibrium ALT beneath the troughs and rims the landscape dynamics reached a new equilibrium state with characteristics that were strongly controlled by the hydrological conditions, in contrast to the initial LCP phase. For the run with $e_{res} = 0.0\,\mathrm{m}$ a HCP landscape was established with water-filled troughs, water-saturated rims,

and relatively dry centres, corresponding to an "inversion" of the hydrological regime during the LCP phase. If the melting of excess ground ice were to continue – for example, induced by a warming climatie – this could possibly lead to the development of a thermokarst lake on longer (multi-decadal to centennial) timescales; this was, however, not observed within the 60-year simulation period. For the run with $e_{res} = -1.0\,\mathrm{m}$ the polygonal landscape stabilized in the ICP phase, with well-drained active layers in all landscape tiles.

The evolution of the polygonal tundra with the phases described above is conceptually depicted in Fig. 15 which has been adapted from Jorgenson et al. (2015). We have demonstrated in this study that the presented model framework can be used to simulate the evolution from un-degraded ice-wedges (with overlying LCP micro-topography), through a phase with initial degradation features (overlying ICP micro-topography), to either advanced degraded ice-wedges (with overlying inundated HCP micro-topography) or interrupted degradation (with and overlying drained ICP micro-topography). However, it is not yet

possible to take into account the feedbacks that lead to long-term (multi-decadal to cenntennial) stabilization and potential aggradation of ice-wedges previously reported by Kanevskiy et al. (2017). The processes involved in these negative feedbacks include the establishment of insulating aquatic vegetation within water-filled troughs and the deposition of laterally transported organic and mineral material above the ice-wedges. The development of thermokarst lakes within the model framework is in theory possible (Westermann et al., 2016; Langer et al., 2016), but would require extended simulation periods and appropri-

ate meteorological forcing. We note that while Jorgenson et al. (2015) and Kanevskiy et al. (2017) developed a qualitative,



conceptual model for the evolution of ice-wedges, our model framework allows a quantitative assessment of the processes and feedbacks involved. It is thus also suitable for the prediction of the future evolution of ice-wedges, which is, however, beyond the scope of this study.

## 5.2 Present-day state of polygonal tundra on Samoylov Island

The ice-wedge polygonal tundra on Samoylov Island is characterized by a large spatial variability, with different types of polygons (low-centred, high-centred) and different moisture levels (drained, inundated). The presence of both water-filled and drained troughs (see Fig. 1, A and C) is indicative of former ice-wedge thermokarst activity. Our model results have improved our understanding of the concurrence of degradation features with un-degraded ice-wedges under the same climatic conditions, by linking it to the spatial variability in site-specific hydrological conditions.

The initial LCP phase, with wet or water-covered centres was associated with the un-degraded LCP micro-topography, which is abundant on the island – particularly in its southern and eastern parts (see Fig. 1, D). Degradation features such as disconnected, water-filled troughs are apparent locally in the central part of the island (see Fig. 1, B). These features are reminiscent of the ICP phase of the run with intermediate hydrological conditions ($e_{res} = 0.0$ m; Fig. 9), during which the ice-wedge melted and the overlying soil layers subsided. Continued ice-wedge degradation in this model run led to collapse of the

rims to levels below those of the centres and inundation of the entire landscape during the HCP phase. This is reminiscent of the advanced degradation features shown in Fig. 1, C. This part of the island currently appears to be very wet and if melting of excess ice continues (as would be particularly likely under a warming climate) then further collapse may occur, leading to the formation of a thermokarst lake. Lastly, drained troughs with LCP and ICP micro-topography are present in the northern part of the island (see Fig. 1, A). These features correspond to the final equilibrium state that is attained in the well-drained run with

$e_{res} = -1.0$ m (Fig. 11), indicating that ice-wedge degradation may have occurred in this part of the island either concurrently with, or prior to, a drainage of the polygonal tundra through the trough network. While Liljedahl et al. (2016) describe this drained ICP/HCP state as representing the final phase of polygonal tundra evolution, in our simulations it is only attained in the runs with draining hydrological conditions. Our numerical modelling results thus correspond well with the conceptual models by Jorgenson et al. (2015) and Kanevskiy et al. (2017), which suggest a number of "pathways" of ice-wedge evolution that are

sometimes cyclic, in contrast to the one-directional evolution suggested by Liljedahl et al. (2016).

  In summary, by making a time-by-space substitution and considering different site-specific hydrological conditions in different parts of the island, our modelling runs have revealed the potential to reproduce a number of observed features of polygonal tundra of Samoylov Island by varying only a single parameter that is reflecting the hydrological conditions in the surroundings of the model domain ($e_{res}$).

In the past few decades, ice-wedge thermokarst has only been locally recorded on Samoylov Island, and at only a moderate rate compared to other sites in the Arctic (Liljedahl et al., 2016). Our modelling results, however, showed rapid ice-wedge degradation under recent climatic forcing for all tested hydrological conditions. This apparent discrepancy may have a number of possible explanations:



- First, ice-wedges on Samoylov Island may have already passed through the rather short transitional phase from the initial LCP state to a new equilibrium state in those parts of the island that were particularly susceptible to ice-wedge degradation in the past. In this case no ice-wedge thermokarst activity would be visible in these parts of the island at present (see the final decade of the model run shown in Fig. 9).

- Second, ice-wedges in some parts of Samoylov Island may still be stable because no initial perturbation of sufficient strength has occurred within the recent past (e.g., due to extreme weather events).

- Third, Liljedahl et al. (2016) detected ice-wedge thermokarst by comparing aerial image time series, which mainly relied on widening of the troughs. The deepening of troughs beneath a water body and the subsidence of rims are, however, not directly detectable using this method.

- Fourth, another possible explanation for the discrepancy is the presence of protective intermediate layers between the active layer and the ice-wedges, as described by Kanevskiy et al. (2017). Although we assumed such a layer in the soil stratigraphy for the trough tiles (see Table 3), it may in reality extend to greater depths, which would lead to a temporal retardation in thermokarst activity.

- Finally, as well as the uncertainty regarding the effect of a protective layer, there are also other processes that can
have a stabilizing effect on ice-wedges which our current model version does not take into account. These include the formation of ground ice and ice-wedge growth, the insulating effects of aquatic vegetation that develops in troughs, and the deposition of organic and mineral material above the ice-wedges, e.g., due to lateral erosion of soil from the rims (see (Abolt et al., 2017) for a modelling approach). The importance of taking into account these stabilizing feedbacks in any long-term numerical modelling that aims to predict the future evolution of polygonal tundra needs to be stressed.

In summary, our modelling results can explain the spatial heterogeneity in the polygonal tundra on Samoylov Island under present-day climatic conditions by relating it to variability in site-specific hydrological conditions. The apparent discrepancy between the ice-wedge degradation simulated by model runs and the low levels of thermokarst activity reported by Liljedahl et al. (2016) can be ascribed to a combination of insufficient in-situ monitoring of ice-wedge thermokarst and an incomplete representation of relevant biogeophysical processes in the model.

**5.3   Possible implications of ice-wedge degradation for ecosystem functions**

The presented model framework makes it possible to quantify changes in the subsurface thermal and hydrological conditions of polygonal tundra in response to the degradation of ice-wedges and consequent changes in the micro-topography. That these changes can be substantial and are strongly dependent on the hydrological conditions, is emphasized by the changes in land-atmosphere and land-land water fluxes (see Fig. 14). The modelling results have shown that the degradation of ice-wedges
increases the susceptibility of polygonal tundra to the hydrology of the surrounding terrain. While the elevated rims in the LCP micro-topography serve as natural barriers to lateral water fluxes into or out of the polygon centres, ice-wedge degradation



leads to subsidence of the rims and thus improved (i.e. occurring earlier in the summer) lateral water flux pathways between polygon centres and the network of troughs.

The increase in total water fluxes (as exemplified by evapotranspiration and runoff) associated with ice-wedge degradation can be interpreted as resulting from an intensification of the water cycling in polygonal tundra, which holds true irrespective
of the hydrological conditions. On a larger scale, such changes in a landscape's water and energy fluxes could induce regional feedbacks to the atmosphere (for example, changes in evapotranspiration affecting cloud formation and hence incoming radiation) and thus ultimately lead to changes in the atmospheric forcing variables that in turn drive the subsurface dynamics (Göckede et al., 2017).

Although our modelling results do not allow us to draw any conclusions regarding the large-scale implications of ice-wedge
degradation, any changes in the subsurface thermal and hydrological regimes of polygonal tundra could result in associated changes to biogeochemical cycling within the active layer, mainly through their control on the decomposition of soil organic carbon (Elberling et al., 2013; Knoblauch et al., 2018; Cresto Aleina et al., 2013; Lara et al., 2015; Grant et al., 2017a). The impacts of global warming on the terrestrial hydrology and ecosystems in the Arctic (e.g. the question of whether ground conditions in certain regions will become dryer or wetter) remain unclear (AMAP, 2017). Our results suggest that permafrost
degradation is characterized by small-scale spatial heterogeneity that may be amplified by increases in thermokarst formation. This makes robust predictions of, for example, the permafrost-carbon feedback, even more difficult.

Our modelling results support the hypothesis that small-scale changes in the micro-topography of ice-rich permafrost landscapes may induce larger-scale feedbacks to the regional ecosystem by altering the water, energy, and carbon fluxes, both within the terrestrial domain and across the surface-atmosphere interface. Since these small-scale features are not taken into
account by the one-dimensional land surface schemes used in climate models, their projections of the future state of permafrost could be biased. Inclusion of small-scale heterogeneity and lateral fluxes in the land surface components of ESMs is therefore highly desirable for any future model developments. The companion paper by Aas et al. (2018) presents a concrete step in this direction.

### 5.4   Advantages over related numerical models of ice-wedge polygonal tundra

A number of numerical modelling approaches have highlighted the important influence exerted by micro-topography and small-scale spatial variability in polygonal tundra on different biogeophysical and biogeochemical processes. The studies by Liljedahl et al. (2016), Bisht et al. (2018) and Abolt et al. (2018) have all identified the influence of spatially heterogeneous snow distributions on subsurface thermal and hydrological regimes and runoff. Kumar et al. (2016), Grant et al. (2017b) and Bisht et al. (2018) also pointed out the crucial influence of lateral subsurface water fluxes on the subsurface thermal state and
active layer thickness. Our model complements and enhances these approaches as it allows investigations to be made into all of the processes and feedbacks mentioned above and is also able to include dynamical topography through its excess ice module (Westermann et al., 2016).

All the above-mentioned investigations have in common that their respective numerical models use two- or three-dimensional spatial domains for subsurface representations. The targeted accuracy of reflecting actual field topographies, however, comes




at a large computational cost, that renders long-term (multi-decadal) simulations unfeasible. Techniques to reduce the spatial dimension of numerical permafrost models are therefore now being developed (Langer et al., 2016; Jan et al., 2018; Aas et al., 2018). The tiling approach of laterally coupled, one-dimensional subsurface representations, that was used in this study, is a trade-off between an accurate reflection of spatial heterogeneity on the one hand (see Sect. 3), and long-term simulations

(covering decades or centuries) on the other hand (see Sect. 4). Our approach is furthermore promising with regard to the up-scaling small-scale processes for inclusion in LSMs, as demonstrated in the companion study by Aas et al. (2018). Because of its independence of lateral scale, the tiling concept is easily transferable to other, in particular larger, landforms, without increasing the computational demands. It could therefore be used to investigate other thermokarst landforms such as lakes (see Langer et al. (2016)), retrogressive thaw slumps, or thermo-erosional valleys, without increasing the computational costs.

Another aspect that distinguishes the model framework presented in this paper from the above-mentioned approaches is the dynamic simulation of changes in micro-topography due to excess ice melt, which is not possible in static terrain representations such as those used in other permafrost models (e.g., Kumar et al. (2016); Bisht et al. (2018); Abolt et al. (2018)). Previous modelling studies have taken into account different polygonal tundra micro-topographies (e.g. low-centred and high-centred polygons) by using separate model runs for different topographies. Our approach, however, allows us to investigate the dy-

namic landscape transition from low-centred polygons to high-centred polygons and the associated transient and asymptotic subsurface thermal and hydrological dynamics. Our results support the findings by Lee et al. (2014) regarding the importance of taking into account excess ground ice in future predictions for permafrost regions. We acknowledge, however, that our model does not take into account other geomorphological processes, such as lateral erosion or sedimentation, in the way that landscape evolution models are able to do (e.g. Coulthard (2001)).

Although the most relevant thermal and hydrological processes, as well as some geomorphological processes, can be taken into account in the CryoGrid3 model, because it is a purely physical model it does not include any parameterization for soil biogeochemistry (such as, for example, in Grant et al. (2017a)), or a dynamic vegetation component such as is included in most LSMs used for large-scale assessments of permafrost regions (e.g., Schaphoff et al. (2013); Chadburn et al. (2015)).

### 5.4.1 Comparison with the companion paper by Aas et al. (2018)

While our study focused on improving our understanding of the processes controlling the evolution of polygonal tundra, a companion paper by Aas et al. (2018) addressed how such small-scale processes could be incorporated into Land Surface Models (LSMs) that can be used for online simulations as part of an Earth System Model (ESM). Our findings complement those of by Aas et al. (2018) who used a similar tiling-approach within the Noah-MP LSM (Niu et al., 2011). They applied their modelling approach to polygonal tundra in the continuous permafrost zone of Northern Siberia as well as to peat plateaus

in the sporadic permafrost zone of Northern Norway. Although the modelling approach of that study has much in common with the approach used in our study (e.g., the tiling concept, the excess ice scheme, and the lateral flux schemes), there are considerable differences between the employed modelling tools, resulting in different, but complementary, objectives of the two papers.





CryoGrid3 is a rather simple LSM, that is dedicated to permafrost applications, and offers a large flexibility, allowing the design of specific model experiments like the three-tile representation of polygonal tundra, coupled to an external water reservoir. Our study using CryoGrid3 hence focused on an improved quantitative understanding of physical permafrost processes on small spatial scales. Our model setup could be easily transferred to other, in particular ice-rich, permafrost landscapes like, for example, peat plateaus in the sporadic permafrost zone.

Noah-MP, on the other hand, is a more sophisticated LSM designed to be used on larger spatial scales, for example, within the scope of numerical weather prediction. The study by Aas et al. (2018) using Noah-MP hence focused on demonstrating the universality of the tiling concept for taking into account small-scale lateral processes in an efficient way, by applying it to two entirely different study areas and landforms.

With respect to polygonal tundra, Aas et al. (2018) were also able to simulate the transition from low-centered polygons to high-centered polygons. In their laterally coupled simulations, however, the polygonal rims are stable under present-day climatic conditions and start to subside around 2030, in a RCP4.5 scenario for Samoylov Island. This difference can likely be attributed to different parameterizations employed in the models, e.g. of the subsurface hydrology. Despite the different timing of excess ice melt, both studies find a similar shift in the patterns of the ground hydrological regimes and surface energy fluxes of polygon centres and rims, induced by the changes in polygon micro-topography.

Taken together, the two papers improved both our understanding of small-scale spatial heterogeneity in permafrost landscapes and the representation of this heterogeneity in LSMs, in a complementary way. While both studies demonstrated the capabilities of the tiling concept, they also shed light on the remaining difficulties of the implementation and the up-scaling of this concept within ESMs, for example, with regard to the representation of site-specific hydrological conditions.

## 6 Conclusions

Our main conclusions from the presented investigations are:

1. A tile-based numerical modelling approach, that takes into account lateral fluxes of heat, water, and snow, is capable of reflecting accurately the spatial heterogeneities in (i) the subsurface thermal and hydrological regimes, (ii) the surface energy balance, and (iii) the snow distribution of polygonal tundra, which are observed in field measurements. In addition, it is capable of simulating the degradation of ice-wedges and the associated changes in micro-topography, from low-centred polygons to high-centred polygons.

2. The timing and speed of ice-wedge degradation is critically affected by the hydrological regime in the active layer beneath the polygon troughs; wetter conditions have a destabilizing effect on ice-wedges and lead to a more rapid degradation than occurs in well-drained landscapes.

3. The spatial variability in the types of ice-wedge polygons observed in the study area (Samoylov Island in the Lena River delta of Northern Siberia) can be linked to the spatial variability in the hydrological conditions.



4. Micro-topographic changes associated with ice-wedge degradation have substantial implications for land-land and land-atmosphere water and energy fluxes, and may also contribute to an intensification of the water cycle in lowland permafrost landscapes.

5. There is a two-way coupling between permafrost hydrology and surface micro-topography, since the hydrological conditions control permafrost degradation and the resulting ground subsidence in turn has a significant effect on the subsurface hydrological regimes.

In summary therefore, our study provides a blueprint for modelling thermokarst landforms and thus helps to disentangle the complex interrelationships between various physical processes shaping ice-rich permafrost landscapes, both in the past and in the future. Together with the companion paper by Aas et al. (2018), this work marks a significant step forward for the representation of small-scale spatial heterogeneity in permafrost landscapes within the land surface schemes of ESMs.

*Code and data availability.* The model for the CryoGrid3 model used for the simulations in this work is available from https://github.com/CryoGrid/CryoGrid3/tree/xice_mpi_polygon_TC. The measurement data from Boike et al. (2018) are available from https://doi.pangaea.de/10.1594/PANGAEA.891142.





## Appendix A: Detailed description of the hydrology scheme

The subsurface hydrology scheme introduced to CryoGrid3 for this study is run at each simulation timestep, given the pre-conditions for infiltration (i.e., no snow cover an unfrozen ground surface). With the introduction of this scheme the (total) volumetric water content ($\theta_w$) of the unfrozen soil cells becomes variable, in contrast to previous versions of CryoGrid3 where

it remained constant. The hydrology scheme computes the changes in water contents due to (i) rainfall, (ii) evapotranspiration, and (iii) infiltration.

*Rainfall:* The rainfall is obtained from the forcing data and is initially put into the uppermost soil grid cell:

$$\delta\theta_{w,P}^1 = \frac{p\Delta t}{\Delta^1} \tag{A1}$$

where $\delta\theta_{w,P}^i$ denotes the change of water content of cell $i$ due to precipitation, $p$ is the precipitation rate ($[\mathrm{m\,s^{-1}}]$), $\Delta t$ is the

timestep ($[\mathrm{s}]$) and $\Delta^i$ the height of grid cell $i$ ($[\mathrm{m}]$).

*Evapotranspiration:* The changes in soil water content resulting from evaporation and transpiration ($\delta w_{ET}^i$) are determined as part of the surface energy balance calculations. The (liquid) water available in the upper part of the soil determines the magnitude of the latent surface heat flux ($Q_e$), which in turn affects the soil water content near the surface.

$Q_e$ is calculated from the individual evaporation and transpiration contributions, but is limited to the potential evaporation

from an unfrozen water surface ($Q_e^*$), which takes into account the atmospheric stability:

$$Q_e = Q_e^* \cdot \min\left[1, \eta_E + \eta_T \frac{r}{1-r}\right] \tag{A2}$$

$$Q_{e,E} = \frac{\eta_E}{\eta_E + \eta_T \frac{r}{1-r}} Q_e \tag{A3}$$

$$Q_{e,T} = \frac{\eta_T}{\eta_E + \eta_T \frac{r}{1-r}} Q_e, \tag{A4}$$

where $\eta_E$ and $\eta_T$ denote the "efficiencies" of evaporation and transpiration, respectively, and $0 \leq r < 1$ is a parameter control-

ling the areal contributions of these processes.

$Q_e^*$ is calculated according to Westermann et al. (2016). The maximum index of the soil cells that are subject to E and T are then calculated as follows:

$$i_E = \min\left(i_{E,max}, i_{AL}\right) \tag{A5}$$

$$i_T = \min\left(i_{T,max}, i_{AL}\right) \tag{A6}$$

where $i_{E,max}$ and $i_{T,max}$ denote the indices of the lowermost grid cells affected by evaporation and transpiration, respectively (corresponding to root depth ($d_T$) and evaporation depth ($d_E$)) and $i_{AL}$ denotes the index of the lowermost cell of the active layer.





With this the overall efficiencies of E and T are calculated as follows:

$$\eta_E = \frac{\sum_{i=1}^{i_E} \sigma^i \Delta^i}{\sum_{i=1}^{i_E} \Delta^i} \tag{A7}$$

$$\eta_T = \frac{\sum_{i=1}^{i_T} \sigma^i \Delta^i}{\sum_{i=1}^{i_T} \Delta^i} \tag{A8}$$

where $\sigma^i = \sigma(\theta_w^i)$ and

$$\sigma(\theta_w) = \begin{cases} 1 & \text{if} \quad \theta_w \geq \theta_{fc} \\ 0.25 \left(1 - \cos\left(\pi \frac{\theta_w}{\theta_{fc}}\right)\right)^2 & \text{if} \quad \theta_w < \theta_{fc} \end{cases} \tag{A9}$$

is a function used to determine the reduction in evaporation and transpiration with decreasing water availability. The same function ($\sigma$) is chosen for E and T, but generally different functions could be used for E and for T.

The water flux associated with $Q_e$ is then uniformly distributed over those parts of the soil that contribute to evaporation and transpiration:

$$\delta\theta_{w,E}^i = -\frac{\sigma^i}{\sum_{i=1}^{i_E} \sigma^i \Delta^i} \frac{Q_{e,E}}{\rho_w L_{lg}} \tag{A10}$$

$$\delta\theta_{w,T}^i = -\frac{\sigma^i}{\sum_{i=1}^{i_T} \sigma^i \Delta^i} \frac{Q_{e,T}}{\rho_w L_{lg}} \tag{A11}$$

$$\delta\theta_{w,ET}^i = \delta\theta_{w,E}^i + \delta\theta_{w,T}^i \tag{A12}$$

where $\sigma^i$ is determined according to Eq. (A9) and $i_E$ and $i_T$ denote the index of the grid cells that coincide with the evaporation depth ($d_E$) and the root depth ($d_T$), respectively.

Note that Equations (A2) and (A10) to (A12) are only used if the surface cell is an unfrozen soil cell. For frozen soil cells a surface resistance to evapotranspiration of $r_s = 50\,\text{s}\,\text{m}^{-1}$ is assumed and the water content remains unchanged. For an unfrozen water surface $r_s = 0$ is used (i.e., $Q_e = Q_{e,pot}$) and the associated change in water content is applied only to the surface cell.

*Infiltration:* After determining the changes in water content due to rainfall ($\delta\theta_{w,P}^1$) and evapotranspiration ($\delta\theta_{w,ET}^i$), water is instantaneously infiltrated into the subsurface. The amount of water per cell in excess of its field capacity ($\theta_{fc}$) is first moved downwards until a frozen cell or the maximum infiltration limit is reached. If there is excess water available the cells are saturated from the bottom upwards, leading to the formation of a water table above the frost table. If there is excess water present after saturating the pore space of the soil, this is either pooled above the soil surface to form a water body, or removed as surface runoff, depending on the model configuration.




## Appendix B: Details on the lateral transport parameterizations

### B1 Lateral transport of heat

The cell-wise effective lateral thermal conductivity ($k^i_{\alpha\beta}$) between tiles $\alpha$ and $\beta$ is calculated from the weighted reciprocal sum of the individual thermal conductivities:

$$k^i_{\alpha\beta} = \frac{A_\alpha + A_\beta}{\frac{A_\alpha}{k^i_\alpha} + \frac{A_\beta}{k^i_\beta}} \ . \tag{B1}$$

### B2 Lateral transport of snow

The redistribution of snow due to wind drift occurs between all tiles of the landscape, irrespective of whether or not they are adjacent. A terrain index ($I_\alpha$) is first calculated for all tiles; it depends on the relative differences between the surface altitudes ($a_\alpha$) at the time of snow transport:

$$\tilde{I}_\alpha = \frac{a_\alpha - \bar{a}}{\sigma_a} \tag{B2}$$

$$I_\alpha = \frac{\tilde{I}_\alpha}{\sum_{\{\alpha|\tilde{I}_\alpha>0\}} \tilde{I}_\alpha} \tag{B3}$$

$$\tag{B4}$$

where $\bar{a} = \sum_\alpha (a_\alpha A_\alpha) / \sum_\alpha A_\alpha$ denotes the area-weighted mean surface altitude, and $\sigma_a$ the area-weighted standard deviation of the surface altitudes of all tiles. Tiles with a positive terrain index ($I_\alpha > 0$) are losing snow, which is then deposited in those tiles that have a negative terrain index ($I_\alpha < 0$). Tiles with $I_\alpha = 0$ have no net change in their snow cover due to lateral transport.

After determining the terrain indices, the volume of drift snow ($V^D$) is accumulated from all tiles with a positive terrain index:

$$V^D = \sum_{\{\alpha|I_\alpha>0\}} \mathrm{SWE}^D_\alpha A_\alpha \tag{B5}$$

where $\mathrm{SWE}^D$ denotes the snow water equivalent that is mobile due to wind drift.

The drift snow is then redistributed between the receiving tiles according to their terrain indices:

$$\delta\mathrm{SWE}_\alpha = \frac{I_\alpha V^D}{A_\alpha} \qquad \forall \alpha \in \{\alpha \mid I_\alpha < 0\} \tag{B6}$$

The snow catch effect of vegetation is taken into account by treating only that part of the snowpack above the maximum vegetation height ($h^{\mathrm{catch}}_\alpha$) as "mobile". Furthermore, lateral snow transport does not occur during melting conditions, i.e., if a cell $i$ of the snowpack has a positive temperature (i.e. $T^i > 0$) or contains liquid water.





## Appendix C: Derivation of topological relationships between the landscape tiles

The topological relationships between the landscape tiles are quantified based on the assumption of a regular hexagonal structure and on estimates of the (typical) polygon size ($A_\text{tot}$) and the relative areal fractions of centres ($\gamma_\text{C}$), rims ($\gamma_\text{R}$), and troughs ($\gamma_\text{T}$).

The areas $A_\alpha$ of the tiles are given as follows:

$$A_\text{C} = \gamma_\text{C} A_\text{tot}, \tag{C1}$$

$$A_\text{R} = \gamma_\text{R} A_\text{tot}, \tag{C2}$$

$$A_\text{T} = \gamma_\text{T} A_\text{tot}. \tag{C3}$$

The lateral geometry of "nested" hexagons is uniquely determined by their area (Fig. C1). The contact lengths between

adjacent ($L$) tiles corresponds to the perimeters of the respective hexagons:

$$L_\text{CR} = 6\sqrt{\frac{2}{3\sqrt{3}} A_\text{C}}, \tag{C4}$$

$$L_\text{RT} = 6\sqrt{\frac{2}{3\sqrt{3}} (A_\text{C} + A_\text{R})}. \tag{C5}$$

The hydraulic distances ($D^\text{hy}$) and thermal distances ($D^\text{th}$) between adjacent tiles need to be specified. For this we considered a one-dimensional cross-section through the hexagonal structures (see the dashed line in Fig. C1 and Fig. C2). Next we

considered the smallest sequence of tiles, that is repeated along this cross-section (see the box in Fig. C2). We assumed this minimal sequence (which is representative for one polygon) to be of length $\sqrt{A_\text{tot}}$, and that the lengths of tiles it consists of to be proportional to their respective areal fraction ($\gamma$). For the hydraulic distances ($D^\text{hy}$) we took the distances between the central point of the rims and the edges of the centres and troughs, thereby assuming a constant hydraulic pressure throughout the centres and troughs. For the thermal distances ($D^\text{th}$) we took the distances between the central points of each pair of adjacent

tiles. Thus we obtained the following relationships for the lateral distances between the tiles:

$$D^\text{th}_\text{CR} = \left(\frac{\gamma_\text{C}}{2} + \frac{\gamma_\text{R}}{4}\right) \sqrt{A_\text{tot}}, \tag{C6}$$

$$D^\text{th}_\text{RT} = \left(\frac{\gamma_\text{T}}{2} + \frac{\gamma_\text{R}}{4}\right) \sqrt{A_\text{tot}}, \tag{C7}$$

$$D^\text{hy}_\text{CR} = \frac{\gamma_\text{R}}{4} \sqrt{A_\text{tot}}, \tag{C8}$$

$$D^\text{hy}_\text{RT} = \frac{\gamma_\text{R}}{4} \sqrt{A_\text{tot}}. \tag{C9}$$

$$\tag{C10}$$

The topological parameter values used for the long-term runs in this study are given in Table C1.





## Appendix D: Parameter overview

An overview of all model parameters of the previous version of CryoGrid3 Xice is provided in Table D1, together with the values used in the present study.

## Appendix E: Modelled and measured active layer temperatures and soil water contents

### E1    Soil temperatures

Figure E1 shows the modelled and measured soil temperatures for a polygon centre (left) at a depth of $0.2\,\mathrm{m}$ and for a polygon rim at a depth of $0.21\,\mathrm{m}$, for 2008 to 2014.

### E2    Soil moisture levels

Figure E2 shows the modelled and measured volumetric soil liquid water contents for a polygon centre (left) at a depth of $0.23\,\mathrm{m}$ and for a polygon rim at a depth of $0.22\,\mathrm{m}$, for 2008 to 2014.



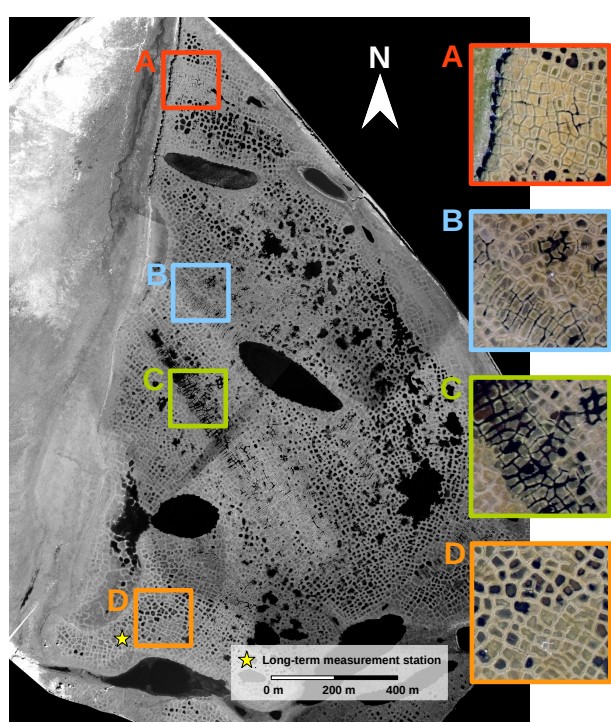

**Figure 1.** Aerial image of Samoylov Island with enlargements showing various types of ice-wedge polygons in different parts of the island, all of which evolved under identical climatic conditions: A - high-centred polygons with drained troughs; B - water-filled troughs that are indicative of initial ice-wedge degradation; C - high-centred polygons with water-filled troughs; D - low-centred polygons with wet or water-covered centres. The central part of the island is relatively wet and contains a large number of water bodies, while the surrounding areas are drier and in part drain into the surrounding river delta. The location of the long-term measurement station for soil and meteorological conditions described in Boike et al. (2018) is indicated by a yellow star.



**A – areal image**   **B – landscape units**   **C – simplified geometry**   **D – tiles**

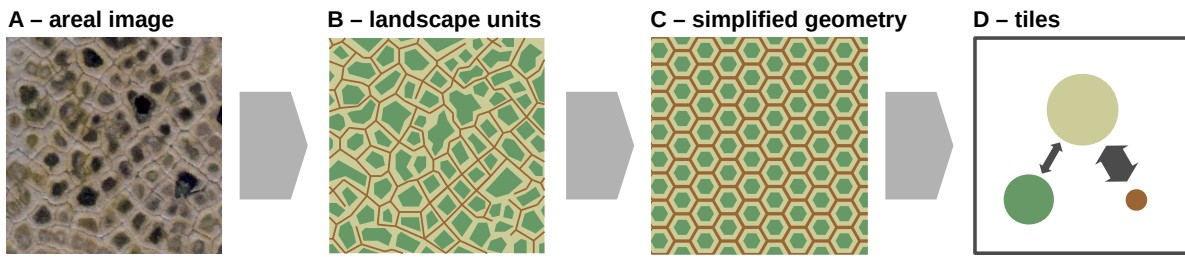

**Figure 2.** In our model framework the spatial heterogeneity of polygonal tundra micro-topography is represented by three landscape tiles (polygon centers (C), polygon rims (R), and troughs (T)). The assumption of equally-sized hexagons arranged on a regular grid makes it possible to quantify lateral transport processes between the tiles.



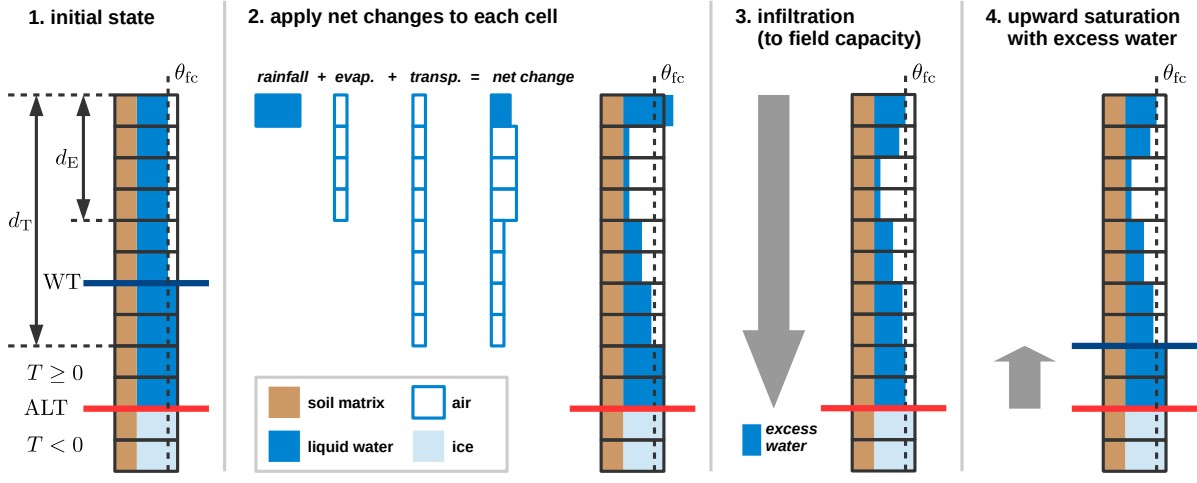

**Figure 3.** Schematic illustration of the hydrology scheme for unfrozen ground conditions. The net changes due to precipitation and evapo-transpiration are calculated during each timestep of the model. The instantaneous infiltration routine involves downwards routing of water to the bottom of the active layer (ALT) and saturation with excess water from the bottom upwards. The water table (WT) forms above the uppermost water-saturated grid cell. Details on the hydrology scheme are given in Appendix A.





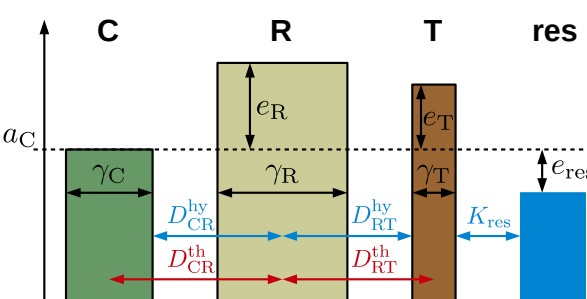

**Figure 4.** Setup of the coupled tiles (centers (C), rims (R), and troughs (T)) with parameters specifying the topology (reflected by areal fractions ($\gamma$), hydrological distances ($D^{\mathrm{hy}}$), and thermal distances ($D^{\mathrm{th}}$)) and micro-topography (reflected by elevations ($e$) relative to the altitude of the centre tile ($a_{\mathrm{C}}$)) of the polygonal tundra landscape. Each tile was assigned a one-dimensional representation of the subsurface, for which a parallelized version of the CryoGrid3 Land Surface Model was used. Water can also be exchanged with the surrounding terrain, which is represented by a theoretical water reservoir (res) with a fixed water level ($e_{\mathrm{res}}$) and a hydraulic conductivity ($K_{\mathrm{res}}$). Subscripts denote the tile(s) the parameters are relating to.





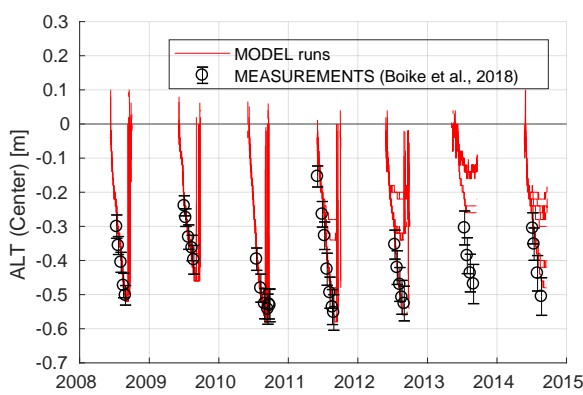 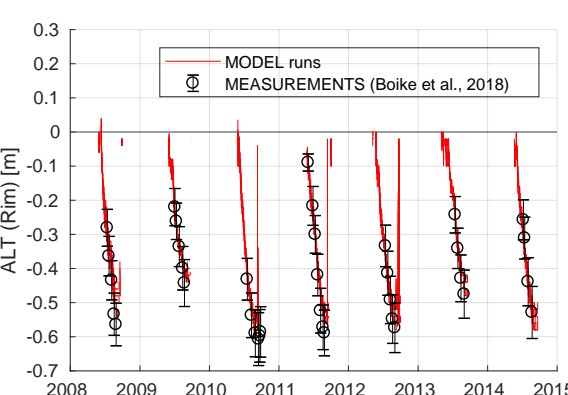

**Figure 5.** Modelled and measured progression of the active layer thicknesses (ALTs) of polygon centres (left panel) and rims (right panel) for the 7-year period of the validation runs. The black markers represent the means and standard deviations of categorized CALM data described in Boike et al. (2018). Red lines represent the individually modelled ALTs in the eight validation runs (Table 4, VALIDATION). Note that the ALTs refer to unfrozen soil, excluding water bodies, so that positive values of ALT may occur if a water body is present.



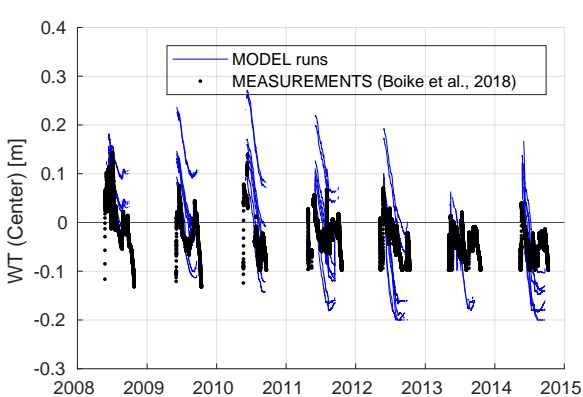

**Figure 6.** Modelled versus measured water tables (WTs) relative to the soil surface of a polygon centre for the 7-year period of the validation runs. The measured data are for a particular polygon centre on Samoylov Island described by Boike et al. (2018). Blue lines represent the individually modelled WTs in the eight validation runs (Table 4, VALIDATION). Note that minimum WT measurements are limited to a level of $-0.120$ m until 2009, and to $-0.095$ m from 2010.



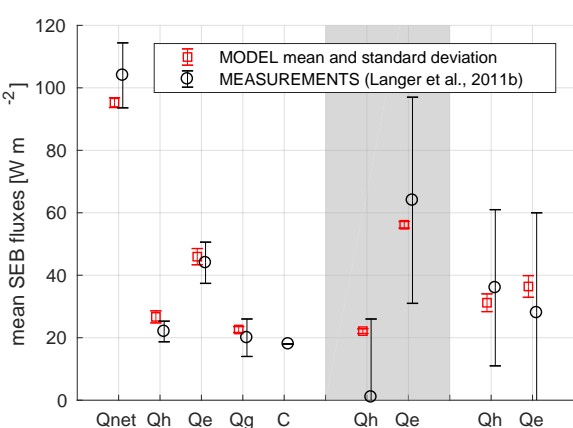

**Figure 7.** Modelled and measured partitioning of the mean surface energy balance (SEB) during summer 2008 (7 Jun - 30 Aug). Model values indicate the mean and standard deviation from all validation runs. The measured data are mean summer fluxes from (Langer et al., 2011b) and the errorbars indicate the estimated accuracies provided in that publication. The left hand side (white background) shows the overall SEB (area-weighed mean of all tiles for the model), the central part of the figure (grey background) shows turbulent fluxes for wet tundra (centre tile for the model), and the right hand side (white background) shows turbulent fluxes for dry tundra (rim tile for the model).





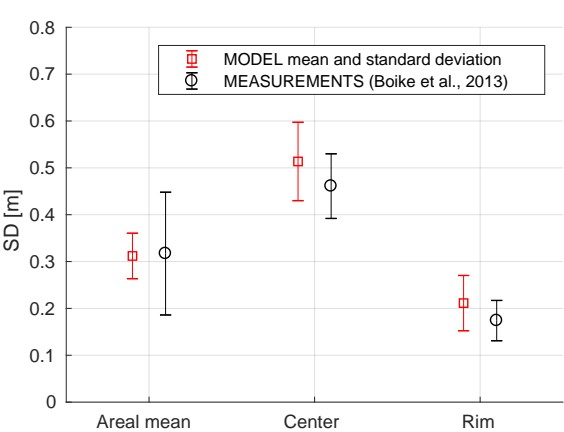

**Figure 8.** Modelled and measured snow depths (SDs) for polygon centres and polygon rims, together with the areal mean. The data from Boike et al. (2013) show the mean and standard deviation of spatially distributed point measurements obtained between 25 April 2008 and 2 May 2008. The model range reflects the standard deviation of the SDs on 1 May 2008 of the eight validation runs (Table 4, VALIDATION).





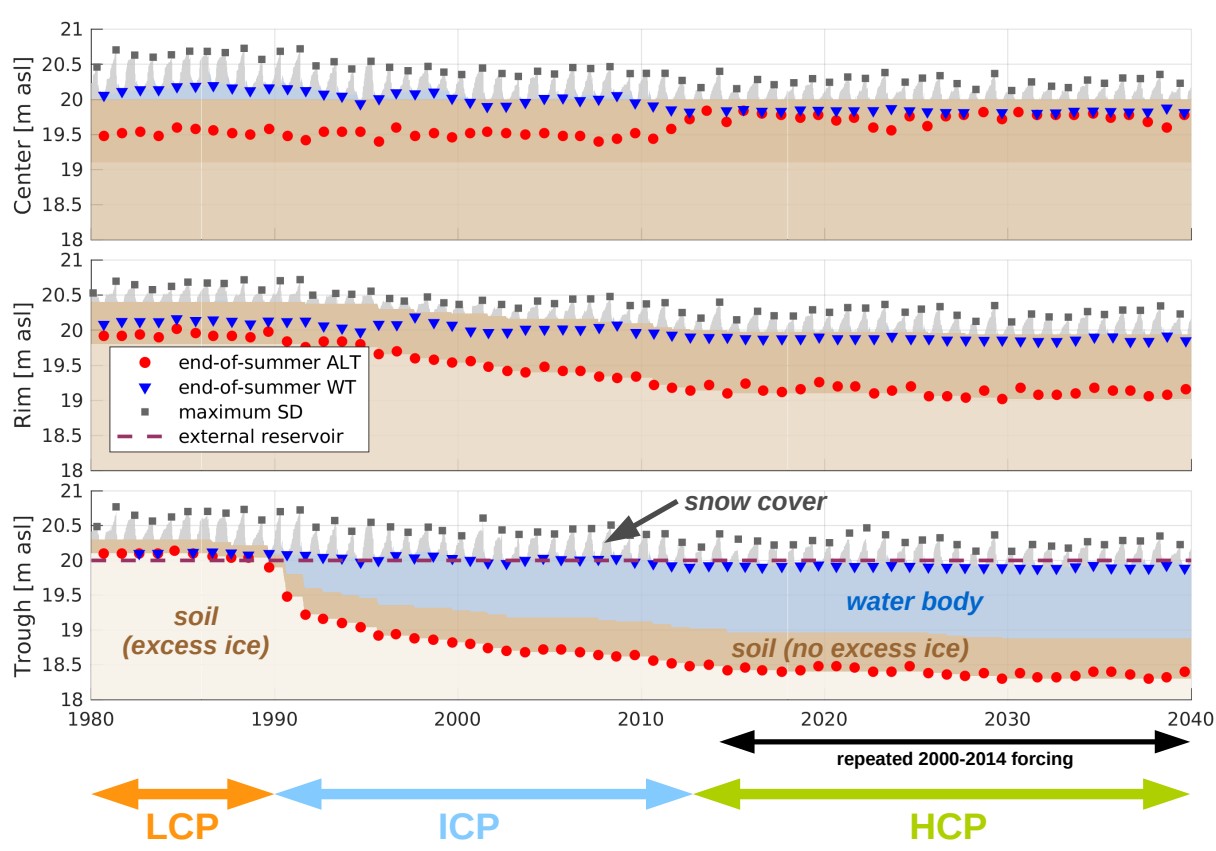

**Figure 9.** Evolution of the polygonal tundra tiles for the 60-year run (from 1980 to 2040) with $e_{res} = 0.0\,\mathrm{m}$ (see Table 4, LONGTERM-XICE). Each panel displays the temporal evolution of the vertical extents of snow cover, water body, and soil domains (excess ice and no excess ice) for the different tiles (top: centre, middle: rim, bottom: trough). For the excess ice layers, lighter colours indicate larger amounts of excess ice. The end-of-summer (maximum) active layer depth (ALT), the end-of-summer water table (WT) and the maximum snow depth (SD) are also indicated for each year. The initial low-centred polygon (LCP) phase lasted for about 11 years, followed by a transitional phase of ice-wedge degradation and ground subsidence (ICP) until the start high-centred polygon (HCP) phase. Note that the meteorological forcing after 2014 consisted of repeated appendments of the forcing between 2000 and 2014. A condensed plot of the results is shown in Fig. 10. The supplementary material to this article contains an animated video showing the results of this simulation run.





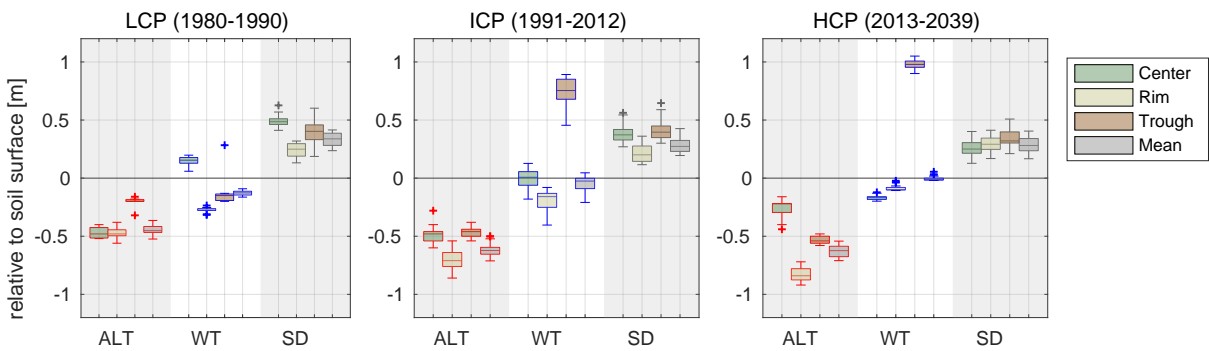

**Figure 10.** Boxplots of the distributions of maximum ALT, WT and SD for each tile and the area-weighted means from all years of the respective phases of the polygonal tundra, from the LCP phase (left panel), through the ICP phase (central panel), to the HCP phase (right panel). The transition from LCP to HCP implies shifts in the thermal and hydrological regimes of the different landscape tiles; it also affects the snow distribution. The results are for the long-term run with $e_{res} = 0.0\,\mathrm{m}$, which is also shown in Fig. 9.





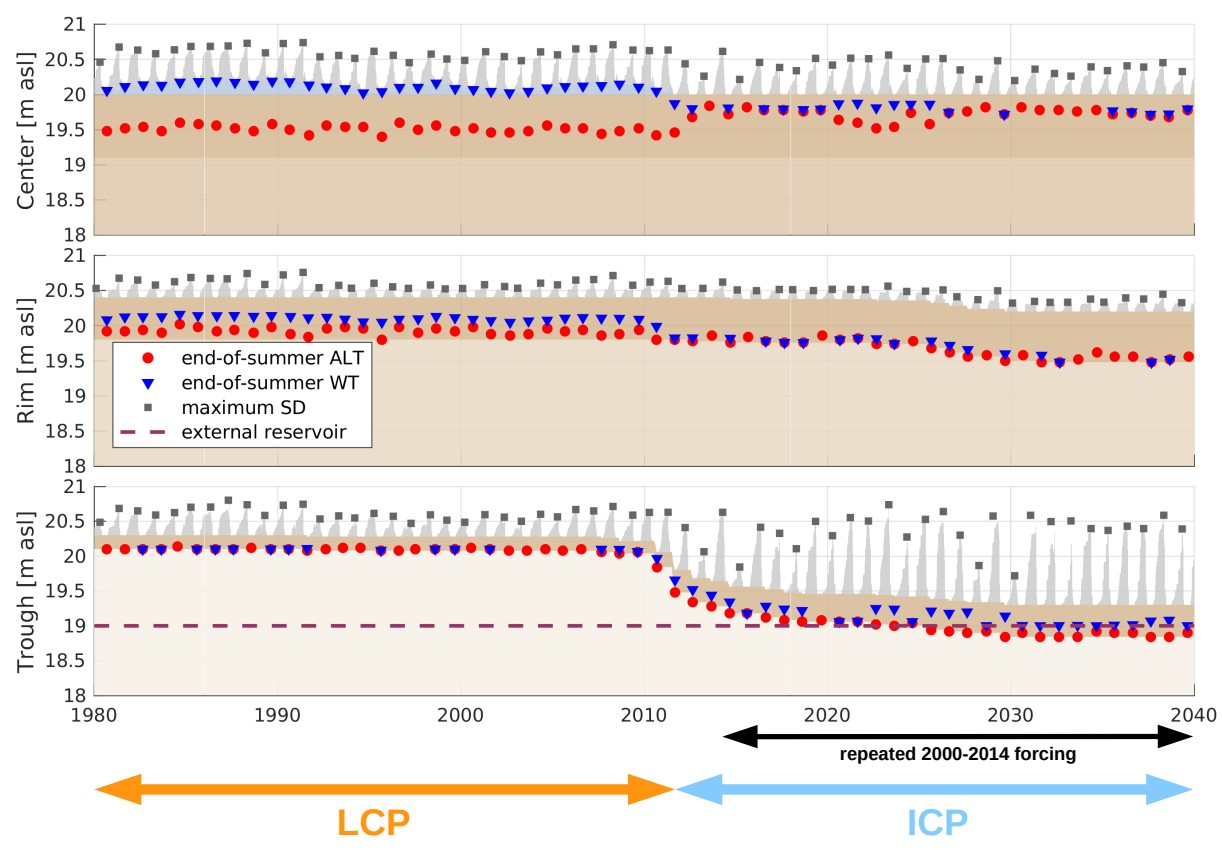

**Figure 11.** Evolution of the polygonal tundra tiles for the 60-year run (from 1980 to 2040) and well-drained hydrological conditions ($e_\mathrm{res} = -1.0\,\mathrm{m}$). Ice-wedge degradation started about two decades later than in the run with a water level in the external reservoir ($e_\mathrm{res} = 0.0\,\mathrm{m}$ – Fig. 9), ultimately leading to an overall lowering of the water tables and effective drainage of the landscape. Note that the meteorological forcing after 2014 consisted of repeated appendments of the forcing between 2000 and 2014. A condensed plot of the results is shown in Fig. 12. The supplementary material to this article contains an animated video showing the results of this simulation run.





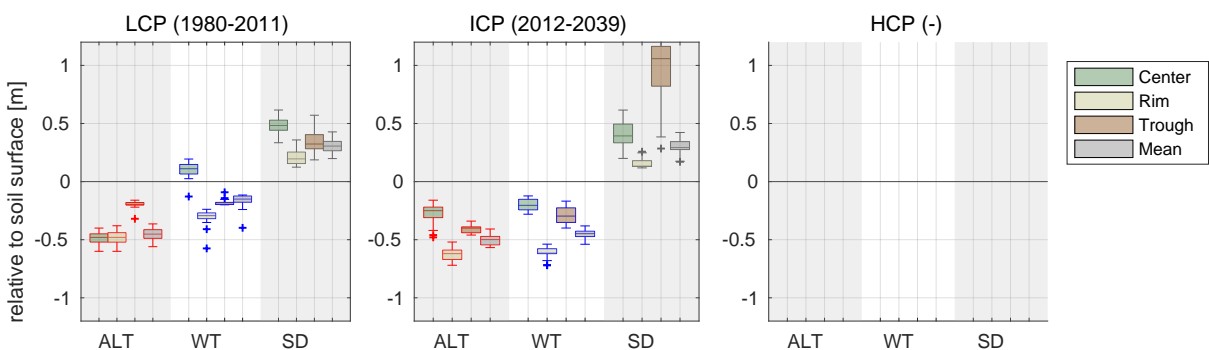

**Figure 12.** Boxplots of the distributions of maximum ALT, WT and SD for each tile and the area-weighted means from all years of the respective phases of polygonal tundra, from the LCP phase (left panel) to the ICP phase (central panel). Note that the HCP phase is not attained during this run. The results are for the long-term run with $e_{res} = -1.0\,\mathrm{m}$, which is also shown in Fig. 11.





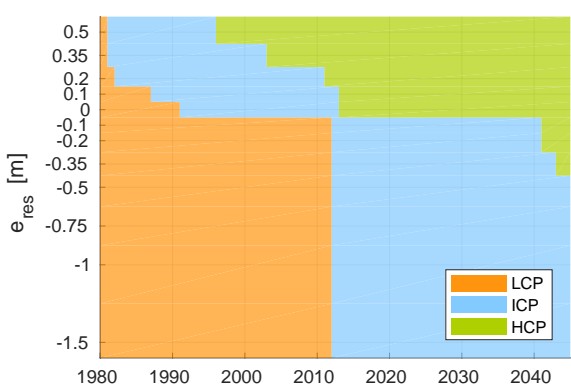

**Figure 13.** The phases of polygonal tundra evolution from low-centred polygons (LCP), through intermediate-centred polygons (ICP), to high-centred polygons (HCP), with respect to the hydrological condition reflected in $e_{res}$. Drainage (low values of $e_{res}$) generally stabilizes the ice-wedges and slows down excess ice melt. Exceptional meteorological conditions can, however, trigger or accelerate ice-wedge degradation (e.g. between 2010 and 2012). The results are for all long-term runs with enabled excess ice module (see Table 4, LONGTERM-XICE).





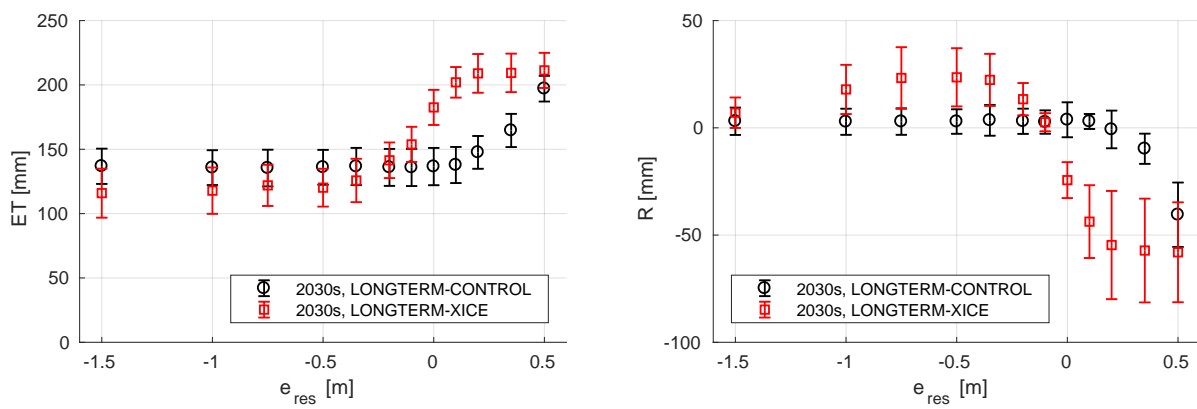

**Figure 14.** Changes to evapotranspiration (E, left panel) and external runoff (R, right panel) induced by the degradation of ice-wedges, for different hydrological conditions. Black markers show the means and standard deviations of the respective fluxes during the final 10 years of the simulation period (2030-2039) ignoring any ground subsidence (i.e., with disabled excess ice module). Red markers show the same for runs with enabled excess ice module, during which ice-wedge degradation occurred within the simulation period.



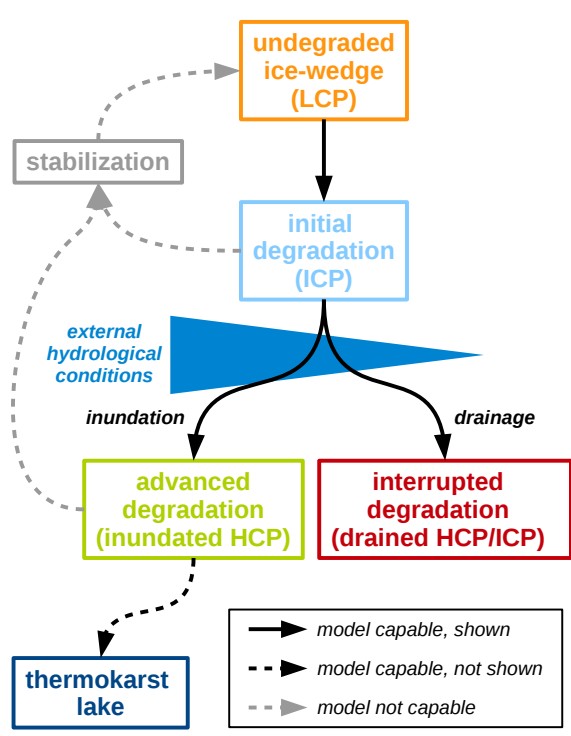

**Figure 15.** Pathways of polygonal tundra landscape evolution, adapted from Jorgenson et al. (2015). While the presented model framework is able to reflect the transition from low-centred polygon terrain to high-centred polygon terrain for various hydrological conditions, it is not able to take into account the long-term stabilization and aggradation of ice-wedges.





**Table 1.** Parameters used to specify the topology and the micro-topography for the tile-based model of polygonal tundra. Note that the initial altitude of the tiles can change due to ground subsidence. Lateral distances and contact lengths are calculated from these values using the formulas in Appendix C.

| Parameter | Symbol | Unit | Center | Rim | Trough | Total | Reference |
|---|---|---|---|---|---|---|---|
| altitude (initial) | $a$ | m | 20 | 20.4 | 20.3 | - | (Boike et al., 2013) |
| area | $A_{\text{tot}}$ | m$^2$ | - | - | - | 140 | (Muster et al., 2012) |
| areal fraction | $\gamma$ | - | 0.3 | 0.6 | 0.1 | 1 | (Muster et al., 2012; Langer et al., 2011b) |





**Table 2.** New parameters introduced with the hydrology scheme and the lateral transport schemes described in this paper.

| Parameter | Symbol | Default value | Unit | Reference |
|---|---|---|---|---|
| *Hydrology scheme* | | | | |
| field capacity / water holding capacity | $\theta_{\mathrm{fc}}$ | 0.50 | - | |
| root depth | $d_{\mathrm{T}}$ | 0.1 (R) | m | (Boike et al., 2018) |
| | | 0.2 (C,T) | m | (Boike et al., 2018) |
| evaporation depth | $d_{\mathrm{E}}$ | 0.1 | m | |
| *Lateral transport schemes* | | | | |
| catch height snow | $h^{\mathrm{catch}}$ | 0.1 | m | (Boike et al., 2018) |
| saturated hydraulic conductivity | $K$ | $1 \cdot 10^{-5}$ | $\mathrm{m\,s^{-1}}$ | (Boike et al., 2018) |
| reservoir hydraulic conductivity | $K_{\mathrm{res}}$ | $5 \cdot 10^{-5}$ | $\mathrm{m\,s^{-1}}$ | (Boike et al., 2018) |
| lateral transport timestep | $\Delta t_{\mathrm{lat}}$ | 0.25 | day | |



**Table 3.** Overview of the soil stratigraphies used for polygon centres, rims, and troughs. Excess ice layers ($\theta_w^0 > \phi_{\text{nat}}$) are shown in bold. Depths are relative to the initial altitude ($a$) of the respective tile.

| Depth [m] | Mineral $\theta_m$ | Organic $\theta_o$ | Nat. porosity $\phi_{\text{nat}}$ | Soil type | Initial water $\theta_w^0$ |
|---|---|---|---|---|---|
| *Center* ($a_{\text{C}} = 20.0$ m) | | | | | |
| 0-0.15 | 0 | 0.15 | 0.85 | sand | 0.85 |
| 0.15-0.30 | 0.20 | 0.05 | 0.75 | sand | 0.75 |
| 0.30-0.90 | 0.30 | 0.05 | 0.65 | silt | 0.65 |
| **0.90-9.00** | **0.30** | **0.05** | **0.55** | **sand** | **0.65** |
| >9.00 | 0.70 | 0 | 0.30 | sand | 0.30 |
| *Rim* ($a_{\text{R}} = 20.4$ m) | | | | | |
| 0-0.10 | 0.10 | 0.15 | 0.75 | sand | 0.50 |
| 0.10-0.60 | 0.30 | 0.05 | 0.65 | silt | 0.65 |
| **0.60-9.40** | **0.20** | **0.05** | **0.55** | **sand** | **0.75** |
| >9.40 | 0.70 | 0 | 0.30 | sand | 0.30 |
| *Trough* ($a_{\text{T}} = 20.3$ m) | | | | | |
| 0-0.20 | 0 | 0.15 | 0.85 | sand | 0.50 |
| **0.20-0.50** | **0.20** | **0.05** | **0.55** | **sand** | **0.75** |
| **0.50-9.30** | **0.05** | **0.05** | **0.55** | **sand** | **0.90** |
| >9.30 | 0.70 | 0 | 0.30 | sand | 0.30 |





**Table 4.** Overview of the configurations used for validation and long-term model runs.

| Run name | Xice | simulation period | areal fractions [-] | | elevations [m] | | snow dens. [$\mathrm{kg\,m^{-3}}$] |
|---|---|---|---|---|---|---|---|
| | | | $\gamma_\mathrm{C}$ | $\gamma_\mathrm{R}$ | $e_\mathrm{R}$ | $e_\mathrm{res}$ | $\rho_\mathrm{snow}$ |
| VALIDATION | off | 10/2007–12/2014 | [0.3, 0.5] | $1 - \gamma_\mathrm{C} - \gamma_\mathrm{T}$ | [0.2, 0.4] | 0.0 | [200, 250] |
| LONGTERM-XICE | on | 10/1979–12/2039 | 0.3 | 0.6 | 0.4 | [-1.5, ... , 0.5] | 200 |
| LONGTERM-CONTROL | off | 10/1979–12/2039 | 0.3 | 0.6 | 0.4 | [-1.5, ... , 0.5] | 200 |





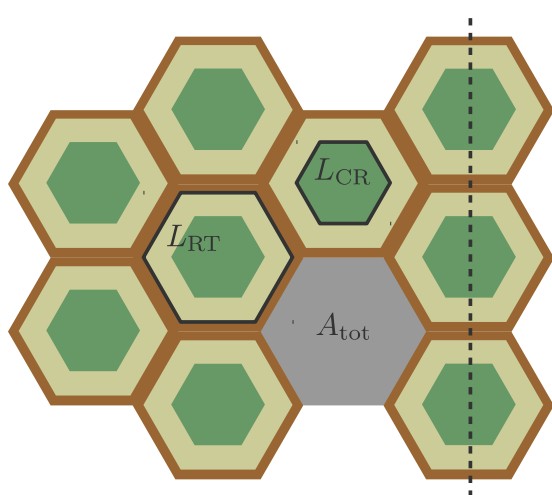

**Figure C1.** Simplified two-dimensional lateral geometry of the polygonal tundra assumed to calculate the contact lengths ($L$) between adjacent tiles.



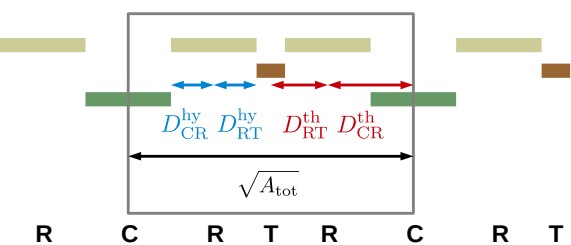

**Figure C2.** Simplified one-dimensional lateral geometry of the polygonal tundra assumed to calculate the distances ($D$) between adjacent tiles.





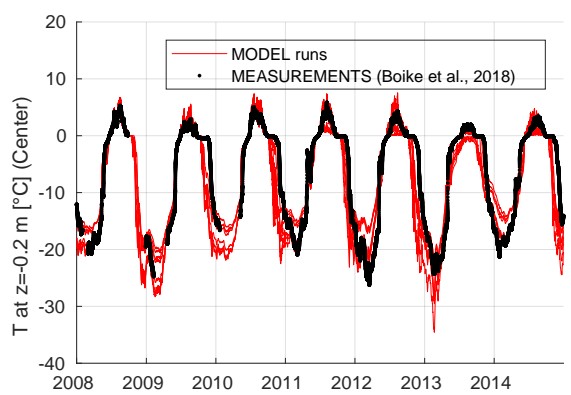 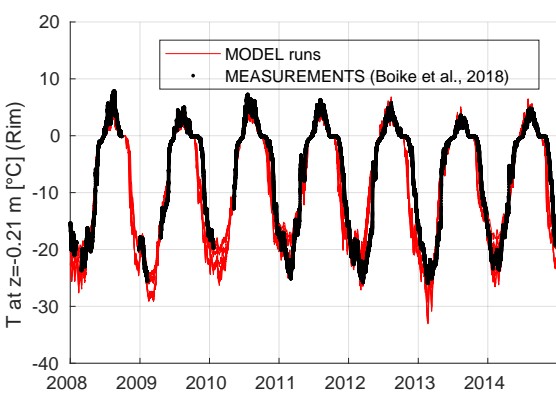

**Figure E1.** Modelled versus measured soil temperatures for a polygon centre (left) at a depth of 0.20 m and a polygon rim (right) at a depth of 0.21 m, for 2008 to 2014. Measurement data from Boike et al. (2018). Model results are for all validation runs (see Table 4, VALIDATION).



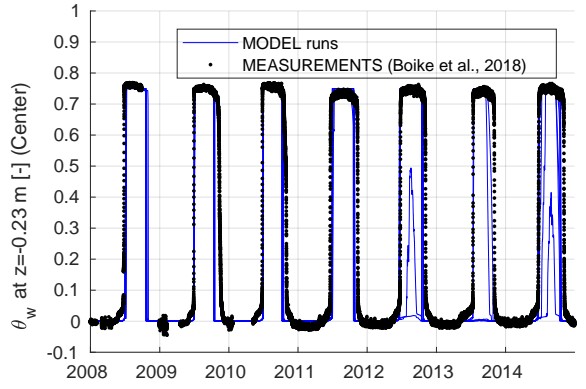
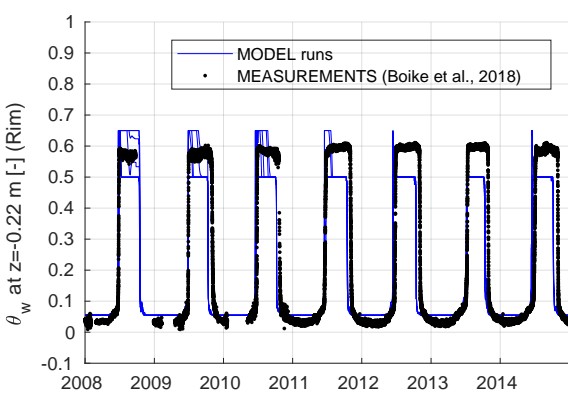

**Figure E2.** Modelled versus measured soil moisture levels for a polygon centre (left) at a depth of $0.23\,\mathrm{m}$ and for a rim (right) at a depth of $0.22\,\mathrm{m}$, for 2008 to 2014. Measurement data from Boike et al. (2018). Model results are for all validation runs (see Table 4, VALIDATION).





**Table C1.** Topological parameter values assumed for the long-term (60-year) runs in this study. The total area of a polygonal structure ($A_{\text{tot}} = 140\,\text{m}^2$) and the areal fractions $\gamma_\alpha$ are based on estimates for Samoylov Island given in Muster et al. (2012). All other topological parameters are calculated from these estimates using the formulas given in Appendix C.

| Parameter | Symbol | Unit | Tiles ($\alpha$, $\alpha\beta$) | | | | |
|---|---|---|---|---|---|---|---|
| | | | C | CR | R | RT | T |
| areal fraction | $\gamma_\alpha$ | - | 0.3 | - | 0.6 | - | 0.1 |
| area | $A_\alpha$ | m$^2$ | 42 | - | 84 | - | 14 |
| contact length | $L_{\alpha\beta}$ | m | - | 24.1 | - | 41.8 | - |
| thermal distance | $D_{\alpha\beta}^{\text{th}}$ | m | - | 3.5 | - | 2.4 | - |
| hydraulic distance | $D_{\alpha\beta}^{\text{hy}}$ | m | - | 1.8 | - | 1.8 | - |





**Table D1.** Parameters used for CryoGrid3 Xice. For all listed parameters we used the same values as documented in Westermann et al. (2016).

| Parameter | Symbol | Default value | Unit |
|---|---|---|---|
| *Surface properties* | | | |
| albedo fresh snow | $\alpha_{\text{snow, max}}$ | 0.85 | - |
| albedo old snow | $\alpha_{\text{snow, min}}$ | 0.50 | - |
| albedo soil | $\alpha_{\text{soil}}$ | 0.20 | - |
| albedo water - unfrozen | $\alpha_{\text{water}}$ | 0.07 | - |
| albedo water - frozen | $\alpha_{\text{ice}}$ | 0.20 | - |
| Time constant of snow albedo change - non-melting | $\tau_{\alpha,\text{f}}$ | 0.008 | $\text{day}^{-1}$ |
| Time constant of snow albedo change - melting | $\tau_{\alpha,\text{m}}$ | 0.24 | $\text{day}^{-1}$ |
| emissivity snow | $\epsilon_{\text{snow}}$ | 0.99 | - |
| emissivity soil | $\epsilon_{\text{soil}}$ | 0.97 | - |
| emissivity water - unfrozen | $\epsilon_{\text{water}}$ | 0.99 | - |
| emissivity water - frozen | $\epsilon_{\text{ice}}$ | 0.98 | - |
| roughness length snow | $z_{0,\text{snow}}$ | 0.0005 | m |
| roughness length soil | $z_{0,\text{soil}}$ | 0.0010 | m |
| roughness length water – unfrozen | $z_{0,\text{water}}$ | 0.0005 | m |
| roughness length water – frozen | $z_{0,\text{ice}}$ | 0.0005 | m |
| resistance against evapostranspiration snow | $r_{\text{s,snow}}$ | 0 | $\text{s}\,\text{m}^{-1}$ |
| resistance against evapostranspiration soil - frozen | $r_{\text{s,frozen soil}}$ | 50 | $\text{s}\,\text{m}^{-1}$ |
| resistance against evapostranspiration water - frozen | $r_{\text{s,ice}}$ | 0 | $\text{s}\,\text{m}^{-1}$ |
| SW radiation extinction coefficient | $\beta_{SW}$ | 25 | $\text{m}^{-1}$ |
| *Material properties* | | | |
| density water | $\rho_{\text{w}}$ | 1000 | $\text{kg}\,\text{m}^{-3}$ |
| density ice | $\rho_{\text{i}}$ | 1000 | $\text{kg}\,\text{m}^{-3}$ |
| density air | $\rho_{\text{a}}$ | 1.293 | $\text{kg}\,\text{m}^{-3}$ |
| volumetric heat capacity water | $C_{\text{w}}$ | $4.2 \cdot 10^6$ | $\text{J}\,\text{K}^{-1}\text{m}^{-3}$ |
| volumetric heat capacity ice | $C_{\text{i}}$ | $1.9 \cdot 10^6$ | $\text{J}\,\text{K}^{-1}\text{m}^{-3}$ |
| volumetric heat capacity air | $C_{\text{a}}$ | $1.3 \cdot 10^3$ | $\text{J}\,\text{K}^{-1}\text{m}^{-3}$ |
| volumetric heat capacity mineral soil | $C_{\text{m}}$ | $2.0 \cdot 10^6$ | $\text{J}\,\text{K}^{-1}\text{m}^{-3}$ |
| volumetric heat capacity organic soil | $C_{\text{o}}$ | $2.5 \cdot 10^6$ | $\text{J}\,\text{K}^{-1}\text{m}^{-3}$ |
| thermal conductivity water | $k_{\text{w}}$ | 0.57 | $\text{W}\,\text{m}^{-1}\,\text{K}^{-1}$ |
| thermal conductivity ice | $k_{\text{i}}$ | 2.20 | $\text{W}\,\text{m}^{-1}\,\text{K}^{-1}$ |
| thermal conductivity mineral soil | $k_{\text{m}}$ | 3.00 | $\text{W}\,\text{m}^{-1}\,\text{K}^{-1}$ |
| thermal conductivity organic soil | $k_{\text{o}}$ | 0.25 | $\text{W}\,\text{m}^{-1}\,\text{K}^{-1}$ |
| latent heat of fusion water | $L_{\text{sl}}$ | $0.334 \cdot 10^6$ | $\text{J}\,\text{kg}^{-1}$ |
| latent heat of vaporization | $L_{\text{lg}}$ | $2.501 \cdot 10^6$ | $\text{J}\,\text{kg}^{-1}$ |
| latent heat of sublimation | $L_{\text{sg}}$ | $2.835 \cdot 10^6$ | $\text{J}\,\text{kg}^{-1}$ |
| *Natural constants* | | | |
| Karman constant | $\kappa$ | 0.4 | - |
| gravitational acceleration | $g$ | 9.81 | $\text{m}\,\text{s}^{-2}$ |
| pressure at sea level | $p_0$ | 100500 | Pa |
| freezing point of water at normal pressure | $T_{\text{f}}$ | 273.15 | K |
| specific gas constant of air | $R$ | 287.058 | $\text{J}\,\text{K}^{-1}\,\text{kg}^{-1}$ |
| Stefan Boltzmann constant | $\sigma$ | $5.6704 \cdot 10^{-8}$ | $\text{W}\,\text{m}^{-2}\text{K}^{-4}$ |
| *Location-specific parameters* | | | |
| geothermal heat flux | $Q_{\text{geo}}$ | 0.05 | $\text{W}\,\text{m}^{-2}$ |





*Author contributions.*   JN designed the study, carried out the simulations and wrote the manuscript. JN and LM implemented the model code. All authors interpreted the results and contributed to the manuscript. SW and JB secured the funding for the project.

*Competing interests.*   JB and ML are members of the editorial board of The Cryosphere. All other authors declare that they have no conflict of interest.

5   *Acknowledgements.*   JN is thankful to the POLMAR graduate school and the Geo.X Young Academy for providing a supportive framework for his PhD project. JN is thankful to Tress Academic for helpful courses on scientific writing and project management. The authors gratefully acknowledge the Climate Geography Group at the Humboldt University of Berlin for providing resources on their high performance computer system. This work was supported by the Research Council of Norway through the PERMANOR project (no. 255331). This work was supported by the Federal Ministry of Education and Research (BMBF) of Germany through a grant to ML (no. 01LN1709A).





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
