# Peer review of "Pathways of ice-wedge degradation in polygonal tundra under different hydrological conditions"

_The Cryosphere, 2018_

## Referee Comment (RC1) · Anonymous Referee #1 · 14 Dec 2018

Nitzbon et al. develop and test a new ice-wedge polygon model to represent thermokarst in polygonal tundra. The paper is very well written, convincingly argued, and balanced. I also think the topic is very important (permafrost degradation) and relevant to ongoing analyses in many groups. I appreciate their creative approach to representing a very spatially heterogeneous system with a geometric scaling approach and their analyses of the sensitivity of their results to the assumptions of their approach.

However, I think the authors could strengthen the paper by considering the following suggestions:

1. The validation runs are worthwhile, and do not appear to result in large discrepancies in moisture or temperature. However, I expect larger variation from uncertainty in soil parameters, and so suggest that such a sensitivity analysis be performed. Pa-

rameters in Tables 2, 3, and D1 are all uncertain, so I would like to see an analysis of which are dominant for the system responses you are studying, and then an uncertainty quantification of your main results associated with variation in the dominant parameters. 2. The hydrology model structure described in Appendix A is somewhat disappointing, given advances made over the past few decades in implementing more sophisticated approaches. However, the proof's in the pudding, and Figure E2 appears to show good comparisons. It is probably worth mentioning in the main text that the model systematically underestimates water content in the rims. That problem may be from setting the porosity to 0.5, but it's not easy to tell. a. Are there no observations at other depths, for both moisture and temperature? Report R2 against observations for temperature and moisture. b. You should describe the model time step and numerical methods for solution. c. Discuss in the main text motivation for your choice of using a simple hydrology model, and what possible implications are. d. Discuss the role of vegetation changes that might be expected during degradation. Currently you set the vegetation parameters at the beginning of the simulation, and I think they remain constant. But, e.g., a drying system should expect to see a transition to plants less adapted to saturated conditions, and that will affect ET. e. It's difficult to see how well the model is doing in Figure 6. Change the y-axis range to -0.2 to 0.3, and report R2 from the average of the 8 simulations, or some combination of those simulations. 3. I am confused about what is being compared in Figure 7. How can an ECOR measurement separate out centers and rims (wet and dry)? Seems impossible, so it's not clear what is being compared. a. Put a 'wet tundra' label above the gray part, and a 'dry tundra' label above the RHS part. And, describe what these terms mean in the context of an ECOR measurement. 4. Line 15 of Page 19, where you use the word "realistic" for your ice-wedge degradation approach. I think you should move the text from lines 12-17 on page 20 up here to show that your results are reasonable, even though you have not made any direct comparisons with degradation. Otherwise, as written, on page 19 I did not see how your representation was reasonable for degradation. 5. In Figure 2, label the colors of the features with a legend.

---

## Referee Comment (RC2) · Anonymous Referee #2 · 24 Jan 2019

This paper presents novel model developments to account for effects of ice-wedge polygon formations in permafrost regions. The scheme is a tile-based approach applied to the CryoGrid3 Land Surface Model which is evaluated against data from a field site in the Lena River delta, and then effects of different hydrological conditions are investigated. The paper is well written, clear, highly relevant and generally complete. However, some details of model need further clarification, and some aspects of the study can be improved, which should only require moderate effort to address.

Clarifications are needed regarding what is actually meant by micro-topography in the context of the model implementation. The tile-based approach, although elegant, is nonetheless representing a multitude of polygons as a single aggregated, effective polygon system. This means many effects of micro topography within the grid cell are

averaged out. This may cause confusion because micro topography effects may be interpreted to refer to the dynamics actually occurring within and between individual polygons in a cluster. This scale of the dynamics does not appear to be represented by the tile-based approach. Also, it is not clear if/how the scheme can be used for clusters with initially mixed HCP/ICP/LCP polygons, which might be relevant for regions where the intended spatial discretization scale of the LSM encompasses two or all three of the polygon categories.

Treatment of water flow is rather simplistic. Groundwater flow is greatly affected by hydraulic conductivity, which is notoriously heterogeneous and varies greatly for different textures and is also challenging to estimate in the field. The value for K shown in Table 2 is quite large and seems to be taken for granted. No uncertainty or error estimate is provided, which is unusual for hydraulic conductivity measurements. A parameter sensitivity study for K would therefore be useful and could provide more insight to the impact of hydrological flows, and in turn, their potential impact on soil T, WT, etc. in the polygon formations.

The assumption of near-instantaneous/rapid vertical water flow might be overly simplistic. (Section 2.2.3, Fig 3). Although hydraulic conductivities of upper soil horizons applicable to the active layer may be high, unsaturated flow is largely dictated by non-linear soil moisture retention curves. Even a small change below saturation can lead to a large decrease in hydraulic conductivity, for some textures by several orders of magnitude, thereby yielding very slow flow rates. Thus, the rapid vertical water flow assumption may be questionable. This may be important because water flow can carry heat through advection, both vertically and laterally, and if near-instantaneous infiltration is assumed, overly non-conservative heat advection may result.

Results, eg Fig 13: The system behavior is greatly controlled by the external boundary condition (the external reservoir). How realistic is this as a BC? A natural hydrological BC is the catchment boundary which would typically be considered no-flow for water, and then internal features such as lakes/reservoirs would be dynamic, resulting from

mass and energy balances including surface and subsurface flows in the catchment. I understand the reservoir concept is used in the study to investigate effects of different hydrological conditions, but it seems this is a somewhat artificial constraint inherent in the model.

Heat flow, Section 2.2.4, Eqn 4 and B1. Eqn B1: How is the thermal conductivity of individual tiles at cells i obtained? (k_alphaˆi and k_betaˆi) Are they a function of the thermal properties of water and ice and soil grains (Table D1, but not apparent in equation B1)?

P11 L19-21: Dry insulating layer – how is this represented in the model? Specifically, how does moisture and air (dryness) influence the thermal properties? I note thermal conductivity of air (or vapor) is not listed in Table D1, nor apparent in eqn B1. See also questions above.

P7 L14-21, Fig 2: Does this mean that all troughs of all polygons are perfectly connected, leading to the same dynamics for all polygons in the grid? Also e.g. Section 2.3.2, Section 2.3.6 – Micro topography is not really represented, it seems all polygons are represented by an single effective polygon-rim-trough system. There is no variability in the dynamics within the multiple and generally diverse polygons as depicted in Fig 2a and 2b.

P11 Section 3.1: Good that full details of the data set is cited but it would help us to know briefly how extensive the data set is, especially how many vertical profiles measuring soil T, moisture, WT etc., exist for each of the different micro-topographic units of the polygonal tundra site. Or is there only one vertical profile per unit type (polygon center, rim, trough)?

Not clear what statistics the error bars represent in e.g. Fig 5 and others. Are these statistics over multiple profiles for each site type, or measurement error, etc.?

P6 Eqn 4: Please clarify what cells i refer to, e.g. vertical discretization.

P7 Eqn 5 and Eqn 7: There is no cells i notation in the Darcy formulations, is this intentional?

P27 Eqn A1: Seems "1" should be cell index "i" to be consistent, typo?

---

## Author Comment (AC1) · 8 Feb 2019

Dear Reviewer,

we thank you very much for your comments which will help to increase the quality of our manuscript. Please find in the following a point-by-point reply to your review. We furthermore provide the revised version of our manuscript as well as an "change-tracked" version in which individual changes with respect to the submitted manuscript are highlighted.

Your comments are in **bold** and extracts from the manuscript are in *italics*, changes to the manuscript are highlighted *yellow*.

> **Nitzbon et al. develop and test a new ice-wedge polygon model to represent thermokarst in polygonal tundra. The paper is very well written, convincingly argued, and balanced. I also think the topic is very important (permafrost degradation) and relevant to ongoing analyses in many groups. I appreciate their creative approach to representing a very spatially heterogeneous system with a geometric scaling approach and their analyses of the sensitivity of their results to the assumptions of their approach.**
> **However, I think the authors could strengthen the paper by considering the following suggestions:**

> **1. The validation runs are worthwhile, and do not appear to result in large discrepancies in moisture or temperature. However, I expect larger variation from uncertainty in soil parameters, and so suggest that such a sensitivity analysis be performed. Parameters in Tables 2, 3, and D1 are all uncertain, so I would like to see an analysis of which are dominant for the system responses you are studying, ...**

We agree that many of the model parameters are uncertain and that it is hence worthwhile to study the sensitivity of the modelled quantities (ALT, WT, SD, SEB, soil temperature, soil moisture) against variations in those parameters. In our initial manuscript we presented a sensitivity analysis for parameters related to the topology, micro-topography and the snow density of our model set-up. However, we just presented the overall spread of all simulations which did not allow to infer the influence of individual parameters. In our revised manuscript we reworked the model validation and sensitivity analysis (section 3). For this, we (i) conducted additional simulations for variations of the field capacity ($\theta_{fc}$) and hydraulic conductivity ($K$), and (ii) compared the modelled quantities with measurements for each parameter variation individually. In contrast to the initial manuscript, we refrained from presenting all possible permutations of the varied parameters, and instead present only runs which differ in one parameter from a set of "default" parameters. The reworked figures (Fig. 5, 6, 7, 8, E1, E2) allow to infer the influence of certain parameters and give an idea of the overall sensitivity of the model to parameter variations. Since the model validation was completely revised, we modified some formulations in sections 3.1 and 3.2 which are highlighted in the "change-tracked" manuscript provided together with the revised manuscript.

> **… and then an uncertainty quantification of your main results associated with variation in the dominant parameters.**

Our main objective was to investigate the influence of different hydrological conditions (reflected in the parameter $e_{res}$) on the degradation of ice-wedges. The focus in this respect was on highlighting the qualitatively different degradation pathways which were simulated within our model framework. While it is likely that other parameters than $e_{res}$ will also influence the timing (and speed) of the degradation, an in-depth analysis of

such factors would go beyond the scope of our study. Hence we did not conduct further "long-term" simulations for our main results section. However, we pointed out the influence of further factors on the modelled ice-wedge degradation in section 5.1 of the discussion:

*It should be noted that – apart from the hydrological conditions reflected in $e_{res}$ – other parameters of the model, including snow properties, the soil stratigraphy, and the depth and amount of excess ice, are likely to affect the timing of the onset of ice-wedge degradation.*

We hope that our revised manuscript (section 3) allows the readers to get a better understanding of the sensitivity of the model to the most uncertain parameters. Given this understanding and the description of the model setup in section 2.3, we are confident that the main results can be interpreted correctly.

**2. The hydrology model structure described in Appendix A is somewhat disappointing, given advances made over the past few decades in implementing more sophisticated approaches. However, the proof's in the pudding, and Figure E2 appears to show good comparisons.**
**It is probably worth mentioning in the main text that the model systematically underestimates water content in the rims. That problem may be from setting the porosity to 0.5, but it's not easy to tell.**

We added the following sentences to section 3.2.2 of the manuscript:

*There was a good agreement between modelled and measured soil moisture levels for the mostly water-saturated center, while the model underestimates soil moisture by about 10% in the dry rim profile. The latter can be attributed to the field capacity parameter ($\theta_{fc}$, Table 2), which is poorly constrained by field measurements.*

**a. Are there no observations at other depths, for both moisture and temperature? Report R2 against observations for temperature and moisture.**

We added comparisons of soil temperature and soil moisture for both polygon centres and troughs at a depth of about 0.40 m to the appendix of the revised manuscript (Figures E1 and E2).

**b. You should describe the model time step and numerical methods for solution.**

We added the following sentence to section 2.2.2 of the manuscript.

*CryoGrid3 uses a first-order forward Euler algorithm with adaptive time step for the numerical integration of the heat conduction equation (see (Westermann et al., 2016) for details).*

**c. Discuss in the main text motivation for your choice of using a simple hydrology model, and what possible implications are.**

We agree that the employed hydrology scheme is rather simplistic and that more sophisticated approaches are available and have already been applied to permafrost settings. However, it constitutes a significant improvement compared to the previous version of CryoGrid3 Westermann et al. (2016), which did not take into account

variable water contents in the active layer at all. The instantaneous infiltration scheme we used, turned out to be sufficiently suited to fulfill the purpose of reflecting the spatial heterogeneity of the ground hydrological regime of polygonal tundra. Moreover, Zhang et al. (2010) showed that instantaneous infiltration algorithms do not necessarily perform worse than more sophisticated schemes. To further justify our methodology, we changed the following formulations in section 2.2.3.

Changed:

*This is a valid assumption for the upper soil layers of tundra wetlands, which are typically characterized by large hydraulic conductivities (Boike et al., 2008) and in which infiltration into the active layer is mainly controlled by thaw depth (Zhang et al., 2010).*

Added:

*We note that the employed hydrology scheme is rather simplistic compared to other schemes available (e.g., Painter et al. (2016)). We confirmed however, that the employed scheme, in combination with the lateral water transport scheme detailed in Sect. 2.2.4, was sufficiently suited to reflect the spatial heterogeneity of the subsurface hydrological regime of polygonal tundra (see Sect. 3).*

**d. Discuss the role of vegetation changes that might be expected during degradation. Currently you set the vegetation parameters at the beginning of the simulation, and I think they remain constant. But, e.g., a drying system should expect to see a transition to plants less adapted to saturated conditions, and that will affect ET.**

It is correct that the vegetation parameters (the root depth $d_R$ and the catch height of snow) remain constatnt throughout the simulations. We agree that the lack of a dynamical vegetation scheme is a limitation of our model, particularly for long-term (decadal or centennial) simulations. We mention this fact in section 5.4 and discussed it further in the revised manuscript.

Added in section 5.4:
*In reality, the change of the subsurface hydrological regimes resulting from ice-wedge degradation (e.g. the drying of polygon centres; Fig. 9), would imply also an adaption of the vegetation (Wolter et al., 2016). This in turn would affect the surface energy balance through changes to the evapotranspiration in a non-trivial way. The development of aquatic vegetation which is also not represented, would have an isolating and thus stabilizing effect on ice-wedges (Kanevskiy et al., 2017).*

**e. It's difficult to see how well the model is doing in Figure 6. Change the y-axis range to -0.2 to 0.3, and report R2 from the average of the 8 simulations, or some combination of those simulations.**

We changed the axis of the figure showing the evolution of WT. Instead of taking an average of the model simulations we display each of the model runs with an individually coloured line and indicate which parameter was varied in each run compared to the default parameter values.

**3. I am confused about what is being compared in Figure 7. How can an ECOR measurement separate out centers and rims (wet and dry)? Seems impossible, so it's not clear what is being compared.**

The eddy-covariance measurements taken from Langer et al. (2011) were conducted at several sites on Samoylov island which had variable areal coverages of wet and dry tundra. Langer et al. (2011) combined these measurements to estimate the contributions of fluxes from wet and dry tundra based on "fractional unmixig". Since this techniques only gives rough estimates, the given uncertainties are accordingly large. We mention this detail about the measurement data in section 3.1 of the revised manuscript:

*These data include a separation of "wet tundra" and "dry tundra" surface energy fluxes which was based on a linear decomposition of measurements conducted in different parts of Samoylov island with variable areal coverages of wet and dry tundra (see Langer et al. (2011b) for details).*

**a. Put a 'wet tundra' label above the gray part, and a 'dry tundra' label above the RHS part. And, describe what these terms mean in the context of an ECOR measurement.**

We added appropriate labels to the figure. For the distinction of wet and dry tundra in the measurements, see answer above.

**4. Line 15 of Page 19, where you use the word "realistic" for your ice-wedge degradation approach. I think you should move the text from lines 12-17 on page 20 up here to show that your results are reasonable, even though you have not made any direct comparisons with degradation. Otherwise, as written, on page 19 I did not see how your representation was reasonable for degradation.**

We agree that the lack of quantitative comparisons with the degradation at our study site does not support the statement made on p.19 l. 15f. With our statement we intended to refer to the reflection of the general process of ice-wedge degradation, independent from our study site. To make this more clear, we moved the comparison with measured subsidence rates to the beginning of the section and modified the original statement about "realistic degradation" such that it refers to the qualitative landscape evolution.

*There is a lack of reliable, long-term measurements of ground subsidence for the different micro-topographic units of polygonal tundra in our study area, which makes quantitative comparisons with the modelled landscape evolution unfeasible. However, Boike et al. (2018) reported recent (2013 to 2017) subsidence rates on Samoylov Island to be in the order of 0.04 m a−1 for polygon rims and < 0.01 m a−1 for polygon centres. These figures are in agreement with the modelled subsidence characteristics, with rates of about 0.02 m a−1 for the rim tiles and no subsidence for the centre tiles (see Figs. 9 and 11). While the modelled ground subsidence seems to be reasonable, the available measurements did not allow for a quantitative comparison of the degradation rates of ice-wedges underneath the troughs. The long-term (60-year) runs with variable hydrological conditions demonstrated, however, that our model framework is able to reflect the process of ice-wedge degradation and the associated changes to the micro-topography of polygonal tundra as described in other studies (e.g., Liljedahl et al. (2016)) in a qualitatively realistic way.*

**5. In Figure 2, label the colors of the features with a legend.**

We changed Figure 2 accordingly.

We hope that we were able to address all questions and comments raised in your review to your satisfaction, and that our revised manuscript is in an adequate state for publication.

Yours sincerely,

Jan Nitzbon (on behalf of the authors)

References

Westermann, S., Langer, M., Boike, J., Heikenfeld, M., Peter, M., Etzelmüller, B., & Krinner, G. (2016). Simulating the thermal regime and thaw processes of ice-rich permafrost ground with the land-surface model CryoGrid 3. *Geosci. Model Dev.*, 9(2), 523–546. https://doi.org/10.5194/gmd-9-523-2016

Zhang, Y., Carey, S. K., Quinton, W. L., Janowicz, J. R., Pomeroy, J. W., & Flerchinger, G. N. (2010). Comparison of algorithms and parameterisations for infiltration into organic-covered permafrost soils. *Hydrology and Earth System Sciences*, 14(5), 729–750. https://doi.org/10.5194/hess-14-729-2010

---

## Author Comment (AC2) · 8 Feb 2019

Dear Reviewer,

we thank you very much for your comments which will help to increase the quality of our manuscript. Please find in the following a point-by-point reply to your review. We furthermore provide the revised version of our manuscript as well as an "change-tracked" version in which individual changes with respect to the submitted manuscript are highlighted.

Your comments are in **bold** and extracts from the manuscript are in *italics*, changes to the manuscript are highlighted *yellow*.

**This paper presents novel model developments to account for effects of ice-wedge polygon formations in permafrost regions. The scheme is a tile-based approach applied to the CryoGrid3 Land Surface Model which is evaluated against data from a field site in the Lena River delta, and then effects of different hydrological conditions are investigated. The paper is well written, clear, highly relevant and generally complete. However, some details of model need further clarification, and some aspects of the study can be improved, which should only require moderate effort to address.**

**Clarifications are needed regarding what is actually meant by micro-topography in the context of the model implementation. The tile-based approach, although elegant, is nonetheless representing a multitude of polygons as a single aggregated, effective polygon system. This means many effects of micro topography within the grid cell are averaged out. This may cause confusion because micro topography effects may be interpreted to refer to the dynamics actually occurring within and between individual polygons in a cluster. This scale of the dynamics does not appear to be represented by the tile-based approach.**

We clarified our notion of "micro-topography" in section 2.2.1 which introduces the tiling approach as referring to the partitioning of polygonal tundra into centres, rims, and troughs:

*We subdivided the polygonal patterned landscape into three landscape units according to what we refer to as its "micro-topography": polygon centres (C), elevated rims (R), and a network of troughs (T) that spreads between the distinct polygonal structures (Fig. 2, B).*

We furthermore stated the approach of a single "effective" polygon being representative for a larger area containing several polygons more precisely.

Removed from section 2.2.1:

**

Added to section 2.2.1:

*Note that apart from the partitioning into centers, rims, and troughs, our approach does not take into account topographic features of individual polygons. Instead, we assumed that larger areas with multiple polygons of similar topography and subject to similar hydrological conditions, can be described via single "effective" polygon composed of the three tiles.*

**Also, it is not clear if/how the scheme can be used for clusters with initially mixed HCP/ICP/LCP polygons, which might be relevant for regions where the intended spatial discretization scale of the LSM encompasses two or all three of the polygon categories.**

As discussed in the article (section 5.4.1) our study aimed at an improved process-understanding and a proof-of-concept for the tiling approach applied to polygonal tundra. In particular, we addressed potential issues of applying it in a straight-forward way to represent entire grid cells of LSMs/ESMs containing polygonal tundra. In the revised manuscript we extended this discussion by pointing also to the problem of various polygon types within one grid cell:

*While both studies demonstrated the capabilities of the tiling concept, they also shed light on the remaining difficulties of the implementation and the up-scaling of this concept within ESMs. The latter comprise the spatial variability of hydrological conditions and the initial presence of different polygon types within one grid cell. Combining the tiling approach with ensembles of simulations might constitute a possibility to bridge this scaling gap.*

**Treatment of water flow is rather simplistic. Groundwater flow is greatly affected by hydraulic conductivity, which is notoriously heterogeneous and varies greatly for different textures and is also challenging to estimate in the field. The value for K shown in Table 2 is quite large and seems to be taken for granted. No uncertainty or error estimate is provided, which is unusual for hydraulic conductivity measurements. A parameter sensitivity study for K would therefore be useful and could provide more insight to the impact of hydrological flows, and in turn, their potential impact on soil T, WT, etc. in the polygon formations.**

In our model the hydraulic conductivity K is a parameter used to quantify the lateral water fluxes between adjacent tiles, according to Eqn. (6). We chose a value which was at the lower end of the values provided in Boike et al. (2018) for the uppermost soil layers, ranging from $1.09*10^{-5}$ m/s to to $46.3*10^{-5}$ m/s. We added the range of values for K to the revised manuscript.

*The saturated soil hydraulic conductivity (K) between all connected tiles was set to $1·10^{-5}$ m s$^{-1}$ and the reservoir hydraulic conductivity ($K_{res}$) was set to $5·10^{-5}$ m s$^{-1}$ both values were of the same order of magnitude as the various estimates for the uppermost soil layers in the same study area ($1.09–46.3 · 10^{-5}$ m s$^{-1}$ ; Boike et al. (2018)).*

We furthermore revised the model validation (section 3) such that the influence of the individually varied parameters becomes apparent. We complemented the parameter variations in the initial manuscript (rim elevation $e_R$, areal fraction center $\gamma_C$, snow density $\rho_{snow}$) by variations of the field capacity ($\theta_{fc}$), and the saturated hydraulic conductivity ($K$). For K we changed the value from the default of $1·10^{-5}$ m s$^{-1}$ to a smaller value of $1·10^{-6}$ m s$^{-1}$. Both the modelled evolutions of ALT and WT turned out to be quite robust against variations of $K$ (see Figures in section 3 of the revised manuscript).

Since the model validation was completely revised, we modified some formulations in sections 3.1 and 3.2 which are highlighted in the "change-tracked" manuscript provided together with the revised manuscript.

**The assumption of near-instantaneous/rapid vertical water flow might be overly simplistic. (Section 2.2.3, Fig 3). Although hydraulic conductivities of upper soil horizons applicable to the active layer may be high, unsaturated flow is largely dictated by non-linear soil moisture retention curves. Even a small change below saturation can lead to a large decrease in hydraulic conductivity, for some textures by several orders of magnitude, thereby yielding very slow flow rates. Thus, the rapid vertical water flow assumption may be questionable. This may be important because water flow can carry heat through advection, both vertically and laterally, and if near-instantaneous infiltration is assumed, overly non-conservative heat advection may result.**

We agree that the employed hydrology scheme assuming instantaneous infiltration is rather simplistic. We are still confident that it was sufficiently suited to fulfill the purpose of reflecting the spatial heterogeneity of the ground hydrological regime of polygonal tundra. Moreover, Zhang et al. (2010) showed that instantaneous infiltration algorithms do not necessarily perform worse than more sophisticated schemes. To stress this limitation of our model we complemented the model description in section 2.2.3 by the following formulations:

*We note that the employed hydrology scheme is rather simplistic compared to other schemes available (e.g., Painter et al. (2016)). However, it constitutes a significant improvement compared to the previous version of CryoGrid3 (Westermann et al., 2016), which did not take into account variable water contents in the active layer at all. We confirmed that the employed scheme, in combination with the lateral water transport scheme detailed in Sect. 2.2.4, was sufficiently suited to reflect the spatial heterogeneity of the subsurface hydrological regime of polygonal tundra (see Sect. 3).*

Regarding the concern about overly non-conservative heat transport, we would like to clarify that in the employed version of CryoGrid3, the process of heat advection is not taken into account. This means that infiltrating water only changes the soil thermal properties and may potentially release latent heat during freezing, but has no direct effect on the temperature of the grid cells. We added the following sentence to section 2.2.3 of the revised manuscript:

*Note that no sensible heat is transported with the infiltrating water, i.e., the process of heat advection is not taken into account by CryoGrid3.*

**Results, eg Fig 13: The system behavior is greatly controlled by the external boundary condition (the external reservoir). How realistic is this as a BC? A natural hydrological BC is the catchment boundary which would typically be considered no-flow for water, and then internal features such as lakes/reservoirs would be dynamic, resulting from mass and energy balances including surface and subsurface flows in the catchment. I understand the reservoir concept is used in the study to investigate effects of different hydrological conditions, but it seems this is a somewhat artificial constraint inherent in the model.**

The external water reservoir used to reflect the site-specific hydrological conditions indeed turned out to exert a strong influence on the system. We used this artificial

boundary condition to reflect contrasting drainage conditions between different parts of our study area (e.g., waterlogged center versus drained margins of Samoylov Island). This boundary condition is a reasonable assumption for our study area (which is mainly flat with no pronounced catchment topography) and objectives, it may be less suited for other study areas, where a coupling to the catchment hydrology would be desirable. We would still like to point out, that via the two parameters (reservoir conductivity and reservoir elevation) the reservoir boundary condition allows a great flexibility. By making the reservoir conductivity dependent on the degree of degradation, one could also reflect the increasing hydrological connectivity of polygonal tundra with ice-wedge degradation. We discussed these limitations in more detail in section 5.4 of the revised manuscript:

*The model setup used in this study makes idealized assumptions on the hydrological connectivity and the hydrological boundary conditions of the polygonal tundra. The connectivity of inter-polygonal troughs which we assumed to be given throughout the simulations, might in reality only develop with advancing degradation of ice-wedges. The assumption of a static external reservoir proved to be useful for comparing contrasting hydrological conditions, but is an idealization which neglects the hydrological dynamics of the surrounding terrain. However, if specific study cases (opposed to our idealized test cases) would require the above-mentioned processes to be taken into account, these could readily be implemented within the CryoGrid3 model framework.*

**Heat flow, Section 2.2.4, Eqn 4 and B1. Eqn B1: How is the thermal conductivity of individual tiles at cells i obtained? (k_alpha^i and k_beta^i) Are they a function of the thermal properties of water and ice and soil grains (Table D1, but not apparent in equation B1)?**

The thermal properties of each soil (and snow) grid cell are calculated based on their composition of their constituents (mineral, organic, water, ice, and air). This is detailed in Westermann et al. (2013), which has been added as a reference. In the revised manuscript we added the heat conduction equation which explicitly contains the thermal properties to the model description and referred to Westermann et al. (2013) for details on the calculation of the thermal properties. This modification should also clarify how the thermal conductivities in Equations 4 and B1 were obtained.

Added to section 2.2.2:

*The numerical model simulates the temporal evolution of the ground temperature profile (T (z)) by solving the one-dimensional heat conduction equation, taking into account the phase change of water through an effective heat capacity:*

$$\left( C(z,T) + \rho_w L_{sl} \frac{\partial \theta_w}{\partial T} \right) \frac{\partial T}{\partial t} = \frac{\partial}{\partial z} \left( k(z,T) \frac{\partial T}{\partial z} \right) \quad ,$$

*where $\theta_w$ is the volumetric water content, $\rho_w$ the density of water, and $L_{sl}$ the latent heat of fusion of water. The thermal properties of the soil cells are (volumetric heat capacity $C(z, T)$ and thermal conductivity $k(z, T)$) are derived from the volumetric fractions of mineral, organic, water, ice, and air (see Westermann et al. (2013) for details).*

**P11 L19-21: Dry insulating layer – how is this represented in the model? Specifically, how does moisture and air (dryness) influence the thermal properties?**

**I note thermal conductivity of air (or vapor) is not listed in Table D1, nor apparent in eqn B1. See also questions above.**

See answer above and Westermann et al. (2013). We added the assumed value for the thermal conductivity of air ($k_a$=0.0243 W/(mK)) to Table D1.

**P7 L14-21, Fig 2: Does this mean that all troughs of all polygons are perfectly connected, leading to the same dynamics for all polygons in the grid? Also e.g. Section 2.3.2, Section 2.3.6 – Micro topography is not really represented, it seems all polygons are represented by an single effective polygon-rim-trough system. There is no variability in the dynamics within the multiple and generally diverse polygons as depicted in Fig 2a and 2b.**

It is correct, that for a simulation with a specified water reservoir, all polygons are represented via an effective polygonal structure, which assumes a perfectly connected network of troughs. A poorly connected network of troughs could be realized by assuming a smaller value for the "reservoir conductivity" $K_{res}$, which was, however, not investigated in this study. For the clarification of our notion of "micro-topography" we refer to our answer to your first point and the respective changes to the manuscript.

**P11 Section 3.1: Good that full details of the data set is cited but it would help us to know briefly how extensive the data set is, especially how many vertical profiles measuring soil T, moisture, WT etc., exist for each of the different micro-topographic units of the polygonal tundra site. Or is there only one vertical profile per unit type (polygon center, rim, trough)?**

The dataset contain one profile of soil temperature and soil moisture for each topographic unit (center, slope, rim, "ice-wedge") and the water table record from one adjacent polygon centre. We added these details to section 3.1:

*This dataset contains vertical soil temperature and soil moisture profiles of different micro-topographic units of the polygonal tundra (one profile for center, slope, rim, and "ice-wedge", respectively; see Fig. 1 for the location of the measurement polygon), as well as water table (WT) records for an adjacent polygon centre.*

**Not clear what statistics the error bars represent in e.g. Fig 5 and others. Are these statistics over multiple profiles for each site type, or measurement error, etc.?**

The error bars in Fig. 5 represent the standard deviation of the active layer depth measurements of the CALM grid on Samoylov for the respective category (39 measurement points for polygon centers, 80 measurement points for polygon rims). We added these details to the manuscript in section 3.1 and the caption of Figure 5.

Added in section 3.1:

*We also used the active layer thickness (ALT) time series from the Samoylov Circumpolar Active Layer Monitoring (CALM) site which cover different micro-topographic units of polygonal tundra, including polygon centres ("wet tundra", n = 39 measurement points) and rims ("dry tundra", n = 80 measurement points) (Boike et al., 2013).*

Caption of Figure 5:

*The black markers represent the means and standard deviations of categorized CALM data (n = 39 measurement points for centers, n = 80 for rims) described in Boike et al. (2018).*

**P6 Eqn 4: Please clarify what cells i refer to, e.g. vertical discretization.**

We changed the formulation in section 2.2.4 to:

*The lateral heat flux between adjacent tiles is computed for each cell of the vertically discretized grid of all tiles, according to Fourier's law. The heat flux $q_{\alpha,i}$ [J s$^{-1}$] to the cell with index i of tile $\alpha$ from all adjacent tiles is given as ...*

We furthermore specified in section 2.2.4 when the lateral heat fluxes are applied:

*The lateral heat fluxes are added after each lateral transport timestep $\Delta t_{lat}$ to the vertical heat fluxes resulting from heat conduction and boundary fluxes (i.e., geothermal and ground heat fluxes).*

**P7 Eqn 5 and Eqn 7: There is no cells i notation in the Darcy formulations, is this intentional?**

As mentioned in Sect. 2.2.4 the lateral water fluxes are calculated as bulk fluxes (rather than cell-wise). They are applied using the instantaneous infiltration scheme which is also used for vertical routing of water. To clarify this, we added the following sentence to section 2.2.4:

*The bulk lateral fluxes $q_{\alpha}$ are applied to each tile $\alpha$ after each lateral transport timestep $\Delta t_{lat}$ using the instantaneous infiltration scheme described in Sect. 2.2.3.*

**P27 Eqn A1: Seems "1" should be cell index "i" to be consistent, typo?**

The "1" refers to the uppermost grid cell of the subsurface which is indexed with 1. We changed the explanation of the symbols in Appendix A so that they refer to this cell rather than to cell "i".

*The rainfall is obtained from the forcing data and is initially put into the uppermost cell (index 1) of the discretized soil grid:*
*...*
*where $\delta\theta_{w,P}^{1}$ denotes the change of water content in the uppermost cell due to precipitation, p is the precipitation rate ([m s$^{-1}$ ]), $\Delta t$ is the timestep ([s]) and $\Delta^{1}$ the height of the uppermost cell ([m]).*

We hope that we were able to address all questions and comments raised in your review to your satisfaction, and that our revised manuscript is in an adequate state for publication.

Yours sincerely,

Jan Nitzbon (on behalf of the authors)

References

Westermann, S., Schuler, T. V., Gisnås, K., & Etzelmüller, B. (2013). Transient thermal modeling of permafrost conditions in Southern Norway. *The Cryosphere, 7*(2), 719–739. https://doi.org/10.5194/tc-7-719-2013

Zhang, Y., Carey, S. K., Quinton, W. L., Janowicz, J. R., Pomeroy, J. W., & Flerchinger, G. N. (2010). Comparison of algorithms and parameterisations for infiltration into organic-covered permafrost soils. *Hydrology and Earth System Sciences, 14*(5), 729–750. https://doi.org/10.5194/hess-14-729-2010

---

## Author Response (AR2)

Dear Christian Beer,

we thank you very much for thoroughly considering the revised version of our manuscript and for providing valuable suggestions for its further improvement. Please find below a point-by-point reply to the remaining points as well as a version of the manuscript with changes highlighted. Your comments are in **bold** and extracts from the manuscript are in *italics*, changes to the manuscript are highlighted *yellow*.

Thank you for submitting a revised version of the manuscript and a detailed 1:1 reply to all reviewer comments. I can see that you addressed almost all reviewer comments. Hence, I am happy to accept this manuscript for publication in TC after some very few minor revisions:

(1) Please, report R2 and RMSE for all comparisons of model results to observations. If there is not enough space available in the figures for that, please use an additional table. Please, also discuss these evaluation metrics in the text.

We provided RMSE, model bias, and R2 values for all model-measurement comparisons in additional tables. Due to space constraints we have put the tables in appendix F of the manuscript, but we discuss them in the main text in sections 3.2.1 and 3.2.2, respectively.

In section 3.2.1, we added:

In addition to the ALT, we also compared modelled and measured soil temperatures within the active layer, for both polygon centres and polygon rims (see Fig. E1 in Appendix E). A detailed assessment of the model performance in terms of root mean squared error (RMSE), bias, and coefficient of determination (R2) is provided in Tabs. F1 and F2 in Appendix F. For the default parameters, there was a slight cold bias ( $\geq -0.4 \text{ °C}$ ) for the simulated rim temperatures while the centre temperatures showed a slight warm bias ( $\leq 0.52 \text{ °C}$ ). R2 values were in the range of 0.69 to 0.83 for the centres, and between 0.79 and 0.90 for the rims, indicating an overall well reproduction of the temperature evolutions.

In section 3.2.2, we added:

The model performance in terms of RMSE, bias, and R2 for the simulated ground hydrological regime, is presented in Tabs. F1 and F3 in Appendix F. The simulated WT had a positive bias (0.06 m) for the default parameters, while it was slightly negative-biased for the runs with  $\gamma C = 0.50$  and eR = 0.20 m. Simulated soil moistures showed mostly low dry-biases ( $\geq -0.08$ ), and fair R2 values ranging from 0.39 to 0.76 for the centres, and from 0.57 to 0.74 for the rims.

Note that we did not provide R2 for the comparisons of ALT and WT since this metric was too sensitive to systematic deviations of ALT or WT in single years, so that the R2 values did not allow insights about the model performance.

(2) Sensitivity study requested by reviewer 1: I agree with you to not perform a full parameter sensitivity study but concentrate on a few important parameters. However, the question was: which parameters (hence processes) are most important for the overall result/conclusion? Please, advance the discussion in that

**respect using the results you presented in the evaluation figures (and R2, RMSE will also support this discussion).**

The sensitivity study presented in the model validation section, hinted at an important influence of the micro-topography of the polygonal tundra (reflected in the elevation of the rims as well as the areal proportions of centres and rims) on simulated ALT, WT and SD in some years. To stress this point, we extended the discussion of the sensitivity analysis in sections 3.2.1, 3.2.2 and 3.3.

In section 3.2.1, we added:

The ALTs of the simulations with an increased areal fraction of the centres ( $\gamma_c = 0.50$ ) and with lower-elevated rims ( $e_R = 0.20$  m) were particularly shallow during the final three years of the validation period, highlighting the sensitivity of the thermal regime to micro-topographic characteristics.

In section 3.2.2, we added:

For some years (e.g., 2011) there were runs in which the centres were water-covered throughout the entire summer while for the runs with  $\gamma_c = 0.50$  and  $e_R = 0.20$  m WT was mostly below the soil surface. This suggests that the simulated WT of the centres is very sensitive to the topology and micro-topography of the polygonal tundra, and feeds back on the simulated ALT as mentioned above.Note that for these runs the RMSE was smaller for the simulated WTs but larger for the simulated ALT of the centres (Tab. F1) compared to the remaining runs, indicating the complex interplay between micro-topography, hydrology, and the active layer.

In section 3.3, we concluded:

The sensitivity tests revealed that the simulated ALT and WT evolutions are robust against variations in snow and hydrological parameters ( $\rho_{snow}$ ,  $\theta_{fc}$ , K), while the polygon micro-topography ( $\gamma_c$ ,  $e_R$ ) had a significant impact on simulated SD, WT and ALT.

We would like to mention, however, that the sensitivity analysis does not allow general conclusions on the effect of the varied parameters on the degradation behaviour of ice-wedges.

Apart from the above-mentioned points we updated two references to discussion papers, which have been published in the meantime:

Aas, K. S., Martin, L., Nitzbon, J., Langer, M., Boike, J., Lee, H., … Westermann, S. (2019). Thaw processes in ice-rich permafrost landscapes represented with laterally coupled tiles in a land surface model. *The Cryosphere*, *13*(2), 591–609. https://doi.org/10.5194/tc-13-591-2019

Boike, J., Nitzbon, J., Anders, K., Grigoriev, M., Bolshiyanov, D., Langer, M., ... Kutzbach, L. (2019). A 16-year record (2002–2017) of permafrost, active-layer, and meteorological conditions at the Samoylov Island Arctic permafrost research site, Lena River delta, northern Siberia: an opportunity to validate remote-sensing data and land surface, snow, and permafrost models. *Earth System Science Data*, *11*(1), 261–299. https://doi.org/10.5194/essd-11-261-2019 We are thankful for the suggestions and confident that the additions and adjustments to our manuscript contribute to its completeness. We hope that we could sufficiently address the remaining points raised by yourself and the reviewers and that our manuscript is now acceptable for publication in TC.

Kind regards,

Jan Nitzbon (on behalf of all authors)

[revised manuscript text omitted]
_{\rm e,T} = \frac{\eta_{\rm T}}{\eta_{\rm E} + \eta_{\rm T} \frac{r}{1-r}} Q_{\rm e},\tag{A4}$$

20 where  $\eta_{\rm E}$  and  $\eta_{\rm T}$  denote the "efficiencies" of evaporation and transpiration, respectively, and  $0 \le r < 1$  is a parameter controlling the areal contributions of these processes.

 $Q_e^*$  is calculated according to Westermann et al. (2016). The maximum index of the soil cells that are subject to E and T are then calculated as follows:

$$i_E = \min\left(i_{\rm E,max}, i_{\rm AL}\right) \tag{A5}$$

$$\quad i_T = \min\left(i_{\text{T,max}}, i_{\text{AL}}\right) \tag{A6}$$

where  $i_{E,max}$  and  $i_{T,max}$  denote the indices of the lowermost grid cells affected by evaporation and transpiration, respectively (corresponding to root depth  $(d_T)$  and evaporation depth  $(d_E)$ ) and  $i_{AL}$  denotes the index of the lowermost cell of the active layer. With this the overall efficiencies of E and T are calculated as follows:

$$\eta_E = \frac{\sum_{i=1}^{i_E} \sigma^i \Delta^i}{\sum_{i=1}^{i_E} \Delta^i}$$
(A7)

$$\eta_T = \frac{\sum_{i=1}^{i_T} \sigma^i \Delta^i}{\sum_{i=1}^{i_T} \Delta^i} \tag{A8}$$

where  $\sigma^i = \sigma(\theta^i_w)$  and

$$\quad \sigma(\theta_w) = \begin{cases} 1 & \text{if} \quad \theta_w \ge \theta_{\text{fc}} \\ 0.25 \left(1 - \cos\left(\pi \frac{\theta_w}{\theta_{\text{fc}}}\right)\right)^2 & \text{if} \quad \theta_w < \theta_{\text{fc}} \end{cases}$$
(A9)

is a function used to determine the reduction in evaporation and transpiration with decreasing water availability. The same function ( $\sigma$ ) is chosen for E and T, but generally different functions could be used for E and for T.

The water flux associated with  $Q_e$  is then uniformly distributed over those parts of the soil that contribute to evaporation and transpiration:

$$\quad \delta\theta^{i}_{w,E} = -\frac{\sigma^{i}}{\sum^{i_{E}} \sigma^{i} \Lambda^{i}} \frac{Q_{e,E}}{\rho_{w} L_{lg}} \tag{A10}$$

$$\delta\theta^{i}_{\mathbf{w},\mathbf{T}} = -\frac{\sigma^{i}}{\sum_{i=1}^{i} \sigma^{i} \Delta^{i}} \frac{Q_{\mathbf{e},\mathbf{T}}}{\rho_{w} L_{lg}} \tag{A11}$$

$$\delta\theta^{i}_{w,ET} = \delta\theta^{i}_{w,E} + \delta\theta^{i}_{w,T} \tag{A12}$$

where  $\sigma^i$  is determined according to Eq. (A9) and  $i_E$  and  $i_T$  denote the index of the grid cells that coincide with the evaporation depth ( $d_E$ ) and the root depth ( $d_T$ ), respectively.

Note that Equations (A2) and (A10) to (A12) are only used if the surface cell is an unfrozen soil cell. For frozen soil cells a surface resistance to evapotranspiration of  $r_s = 50 \text{ sm}^{-1}$  is assumed and the water content remains unchanged. For an unfrozen water surface  $r_s = 0$  is used (i.e.,  $Q_e = Q_{e,pot}$ ) and the associated change in water content is applied only to the surface cell.

*Infiltration:* After determining the changes in water content due to rainfall  $(\delta \theta_{w,P}^1)$  and evapotranspiration  $(\delta \theta_{w,ET}^i)$ , water is instantaneously infiltrated into the subsurface. The amount of water per cell in excess of its field capacity  $(\theta_{fc})$  is first moved

20 downwards until a frozen cell or the maximum infiltration limit is reached. If there is excess water available the cells are saturated from the bottom upwards, leading to the formation of a water table above the frost table. If there is excess water present after saturating the pore space of the soil, this is either pooled above the soil surface to form a water body, or removed as surface runoff, depending on the model configuration.

**Appendix B: Details on the lateral transport parameterizationsschemes**

**B1** Lateral transport of heat**

The cell-wise effective lateral thermal conductivity  $(k_{\alpha\beta}^i)$  between tiles  $\alpha$  and  $\beta$  is calculated from the weighted reciprocal sum of the individual thermal conductivities:

5
$$k_{\alpha\beta}^{i} = \frac{A_{\alpha} + A_{\beta}}{\frac{A_{\alpha}}{k_{\alpha}^{i}} + \frac{A_{\beta}}{k_{\beta}^{i}}}.$$
 (B1)

**B2** Lateral transport of snow**

The redistribution of snow due to wind drift occurs between all tiles of the landscape, irrespective of whether or not they are adjacent. A terrain index ( $I_{\alpha}$ ) is first calculated for all tiles; it depends on the relative differences between the surface altitudes ( $a_{\alpha}$ ) at the time of snow transport:

10
$$\tilde{I}_{\alpha} = \frac{a_{\alpha} - \bar{a}}{\sigma_{\alpha}}$$
 (B2)

$$I_{\alpha} = \frac{\tilde{I}_{\alpha}}{\sum_{\left\{\alpha \mid \tilde{I}_{\alpha} > 0\right\}} \tilde{I}_{\alpha}}$$
(B3)

(B4)

where ā = Σα (aαAα)/ΣαAα denotes the area-weighted mean surface altitude, and σa the area-weighted standard deviation of the surface altitudes of all tiles. Tiles with a positive terrain index (Iα > 0) are losing snow, which is then deposited in
those tiles that have a negative terrain index (Iα < 0). Tiles with Iα = 0 have no net change in their snow cover due to lateral transport.

After determining the terrain indices, the volume of drift snow  $(V^{D})$  is accumulated from all tiles with a positive terrain index:

$$V^{\rm D} = \sum_{\{\alpha | I_{\alpha} > 0\}} \text{SWE}^{\rm D}_{\alpha} A_{\alpha}$$
(B5)

20 where  $SWE^{D}$  denotes the snow water equivalent that is mobile due to wind drift.

The drift snow is then redistributed between the receiving tiles according to their terrain indices:

$$\delta SWE_{\alpha} = \frac{I_{\alpha}V^{D}}{A_{\alpha}} \qquad \forall \alpha \in \{\alpha \mid I_{\alpha} < 0\}$$
(B6)

The snow catch effect of vegetation is taken into account by treating only that part of the snowpack above the maximum vegetation height ( $h_{\alpha}^{\text{catch}}$ ) as "mobile". Furthermore, lateral snow transport does not occur during melting conditions, i.e., if a cell *i* of the snowpack has a positive temperature (i.e.  $T^i > 0$ ) or contains liquid water.

**Appendix C: Derivation of topological relationships between the landscape tiles**

The topological relationships between the landscape tiles are quantified based on the assumption of a regular hexagonal structure and on estimates of the (typical) polygon size ( $A_{tot}$ ) and the relative areal fractions of centres ( $\gamma_C$ ), rims ( $\gamma_R$ ), and troughs ( $\gamma_T$ ).

5 The areas  $A_{\alpha}$  of the tiles are given as follows:

$$A_{\rm C} = \gamma_{\rm C} A_{\rm tot},\tag{C1}$$

$$A_{\rm R} = \gamma_{\rm R} A_{\rm tot},\tag{C2}$$

$$A_{\rm T} = \gamma_{\rm T} A_{\rm tot}.$$
 (C3)

The lateral geometry of "nested" hexagons is uniquely determined by their area (Fig. C1). The contact lengths between 10 adjacent (L) tiles corresponds to the perimeters of the respective hexagons:

$$L_{\rm CR} = 6\sqrt{\frac{2}{3\sqrt{3}}A_{\rm C}},\tag{C4}$$

$$L_{\rm RT} = 6\sqrt{\frac{2}{3\sqrt{3}}} \left(A_{\rm C} + A_{\rm R}\right).$$
(C5)

The hydraulic distances  $(D^{hy})$  and thermal distances  $(D^{th})$  between adjacent tiles need to be specified. For this we considered a one-dimensional cross-section through the hexagonal structures (see the dashed line in Fig. C1 and Fig. C2). Next we considered the smallest sequence of tiles, that is repeated along this cross-section (see the box in Fig. C2). We assumed this minimal sequence (which is representative for one polygon) to be of length  $\sqrt{A_{tot}}$ , and that the lengths of tiles it consists of to be proportional to their respective areal fraction ( $\gamma$ ). For the hydraulic distances ( $D^{hy}$ ) we took the distances between the central point of the rims and the edges of the centres and troughs, thereby assuming a constant hydraulic pressure throughout the centres and troughs. For the thermal distances ( $D^{th}$ ) we took the distances between the central points of each pair of adjacent

20 tiles. Thus we obtained the following relationships for the lateral distances between the tiles:

$$D_{\rm CR}^{\rm th} = \left(\frac{\gamma_{\rm C}}{2} + \frac{\gamma_{\rm R}}{4}\right)\sqrt{A_{\rm tot}},\tag{C6}$$

$$D_{\rm RT}^{\rm ln} = \left(\frac{\gamma_{\rm I}}{2} + \frac{\gamma_{\rm K}}{4}\right) \sqrt{A_{\rm tot}},\tag{C7}$$
$$D_{\rm hy}^{\rm hy} = \frac{\gamma_{\rm R}}{\sqrt{A_{\rm tot}}},\tag{C8}$$

$$D_{\rm CR} = \frac{1}{4} \sqrt{A_{\rm tot}},$$

$$(Co)$$

$$D_{\rm RT}^{\rm hy} = \frac{\gamma_{\rm R}}{4} \sqrt{A_{\rm tot}}.$$
(C9)

(C10)

25

The topological parameter values used for the long-term runs in this study are given in Table C1.

**Appendix D: Parameter overview**

An overview of all model parameters of the previous version of CryoGrid3 Xice is provided in Table D1, together with the values used in the present study.

**Appendix E: Modelled and measured active layer temperatures and soil water contents**

**5 E1 Soil temperatures**

Figure E1 shows the modelled and measured soil temperatures (T) for a polygon centre (left) at a depth of 0.2 m and at depths of 0.20 m and 0.40 m, and for a polygon rim at a depth depths of 0.21 m, for and 0.38 m, during the 7-year validation period from 2008 to 2014.

**E2 Soil moisture levels**

10 Figure E2 shows the modelled and measured volumetric soil liquid water contents  $(\theta_w)$  for a polygon centre (left) at a depth at depths of 0.23 m and 0.43 m, and for a polygon rim at a depth depths of 0.22 m, for and 0.37 m, during the 7-year validation period from 2008 to 2014.

**Appendix F: Model performance assessment**

We assessed the ability of the model to reproduce measurements of ALT, WT, soil temperatures and soil moisture levels by

15 calculating the root mean squared error (RMSE), the model bias, and the coefficient of determination  $(R^2)$  for all validation runs. The respective scores are provided in Tab. F1 for ALT and WT, in Tab. F2 for soil temperatures, and in Tab. F3 for soil moisture levels.